# Thermodynamics and kinetics guided probe design for uniformly sensitive and specific DNA hybridization without optimization

Xin Chen [1,4], Na Liu [1,4], Liquan Liu [1,4], Wei Chen[2], Na Chen[1], Meng Lin[1], Jiaju Xu [1], Xing Zhou[3], Hongbo Wang[3]*, Meiping Zhao [2]* & Xianjin Xiao [1,3]*

Sensitive and specific DNA hybridization is essential for nucleic acid chemistry. Competitive composition of probe and blocker has been the most adopted probe design for its relatively high sensitivity and specificity. However, the sensitivity and specificity were inversely correlated over the length and concentration of the blocker strand, making the optimization process cumbersome. Herein, we construct a theoretical model for competitive DNA hybridization, which disclose that both the thermodynamics and kinetics contribute to the inverse correlation. Guided by this, we invent the 4-way Strand Exchange LEd Competitive DNA Testing (SELECT) system, which breaks up the inverse correlation. Using SELECT, we identified 16 hot-pot mutations in human genome under uniform conditions, without optimization at all. The specificities were all above 140. As a demonstration of the clinical practicability, we develop probe systems that detect mutations in human genomic DNA extracted from ovarian cancer patients with a detection limit of 0.1%.

[1] Institute of Reproductive Health, Tongji Medical College, Huazhong University of Science and Technology, Wuhan 430030, PR China. [2] Beijing National Laboratory for Molecular Sciences, MOE Key Laboratory of Bioorganic Chemistry and Molecular Engineering, College of Chemistry and Molecular Engineering, Peking University, Beijing 100871, PR China. [3] Department of Obstetrics and Gynaecology, Union Hospital, Tongji Medical College, Huazhong University of Science and Technology, Wuhan 430022, PR China. [4] These authors contributed equally: Xin Chen, Na Liu, Liquan Liu. *email: hb_wang1969@sina.com; mpzhao@pku.edu.cn; xiaoxianjin@hust.edu.cn

Sensitive and specific DNA hybridization has long been the essence and foundation of DNA based sensors, assays and materials. Especially for the detection of small genetic variations, which is the core step of various molecular diagnostic assays, such as genotyping[1–3], DNA microarray[4–6] and mutation detection[7–9], requires the method to be highly sensitive and specific. In particular, the circulating tumour DNA (ctDNA)[8,10–15], which is currently the most promising biomarker for early diagnosis of cancer[11,12,14,16], is submerged in large excess of wild-type cell free DNA[15,17]. Thus, detection of ctDNA requires the method to be highly discriminative towards mutant-type DNA and wild-type DNA, which is only single-base different between each other[1,10,11,15].

Numerous nucleic acid assays have been developed for discrimination of single-base mismatches including probes with complex structures[18,19], enzyme assisted DNA probes[20,21], barcode assays[22,23], selective PCR[24,25] and next-generation sequencing[26,27]. Principally, all of these methods rely on the specificity of DNA hybridization at some steps of their working procedures. However, even under optimized conditions, discrimination of single-base mismatches is challenging, especially for stable mismatches such as X:G (X = A,T,G) which cause considerably slighter changes to the thermodynamics as compared with other types of mismatches such as X:C[28–31], leading to low discrimination efficiency.

Competitive DNA strands are commonly used to enhance the discrimination efficiency of the DNA probe system[32–34]. They are termed as Blocker or Clamp in different occasions. The blocker strand would preferably hybridize with wild-type DNA and prevent it from further hybridizing with probes. Consequently, the blocker strand enriched the percentage of mutant-type DNA in the pool of free ssDNA, which facilitated the subsequent discrimination process of the probe. Crucially, competitive composition of probe and blocker has been the most adopted probe design in nucleic acid assays.

However, as reported in previous literatures, the sensitivity of the competitive composition systems was in reverse correlation to the specificity when changing the length and concentration of blocker, which meant higher sensitivity led to lower specificity and vice versa[32]. This intrinsic contradiction resulted in labour intensive optimization of the structure and concentration of the blocker strand and probe. Zhang et al. established a simulation guided kinetic model to algorithmize the optimization process and help researchers to compute the ideal sequence and structure of the probe and blocker strand[32]. The prediction was relatively accurate and greatly facilitated the applications of the competitive probe systems. Nevertheless, the established model was still subjected to the intrinsic contradiction, and the optimized sensitivity and specificity were trade-off results, both of which failed to reach the theoretical maximum value. Aside from the loss of detection performance, the design process was still complicated, as experimental optimization and verification was further needed to assure the assay performance. Thus, high-throughput analysis of multiple mutations[35–37], which required design of multiple compositions of blocker strand and probe, was very difficult and time-consuming under current DNA probe systems.

Herein, we have constructed a complete theoretical model for competitive DNA hybridization. The model described the variation of sensitivity and specificity over the length and concentration of the blocker strand. Through the model, we disclosed that it was the inverse correlation in thermodynamics and the kinetic traps that led to the inverse correlation between sensitivity and specificity of the competitive probe/blocker systems. Guided by the thermodynamics and kinetics, we invented a novel probe/blocker system based on Holliday junction branch migration: the 4-way Strand Exchange LEd Competitive DNA Testing system

(4-way SELECT system). Both the theoretical calculations and experimental results demonstrated that the novel system broke up the intrinsic inverse correlation: as the length and concentration of blocker strand increased, the sensitivity remained unchanged while the specificity increased monotonically until reaching to the theoretical maximum value. Therefore, the sequence design and reaction conditions can be simplified and uniformed without any optimization at all, which could greatly facilitate and broaden the application range of DNA probe, and in particular, held great potential for development of high-throughput analysis of multiple mutations.

## Results

**Conventional probe/blocker composition system.** We first wanted to construct a thermodynamic model describing the sensitivity and specificity of conventional probe/blocker composition system over the length and concentration of blocker strand. As shown in Fig. 1a, in a standard probe/standard blocker composition system, the targeting strand MT was perfectly matched to the probe and single-base mismatched to the blocker, and on the contrary, the interfering strand WT was single-base mismatched to the probe and perfectly matched to the blocker. The ultimate goal of a probe/blocker composition system was to differentiate the targeted MT strands from the interfering WT strands. Therefore, sensitivity and specificity were the most important parameters to assess its functionality. The sensitivity and specificity were defined as follows:

$$\text{Sensitivity} = \frac{[\text{PMT}]}{c_0} \tag{1}$$

$$\text{Specificity} = \frac{[\text{PMT}]}{[\text{PWT}]} \tag{2}$$

where, $c_0$ was the initial concentration of MT or WT. Adopting the mass-action equilibrium model and appropriate approximations, we could obtain the analytic expressions of sensitivity and specificity over $[\text{B}]_0$ and $-\Delta G_{\text{BW}}$ (see Supplementary Note 1 for detailed calculation),

$$\text{Sensitivity}\big([\text{B}]_0, -\Delta G_{\text{BW}}\big) = \frac{e^{-\frac{\Delta G_{\text{PM}}}{RT}}[\text{P}]_0}{e^{-\frac{\Delta G_{\text{BW}}+\Delta\Delta G_{\text{B}}}{RT}}[\text{B}]_0 + e^{-\frac{\Delta G_{\text{PM}}}{RT}}[\text{P}]_0 + 1}$$
$$\in \left[\frac{e^{-\frac{\Delta G_{\text{PM}}}{RT}}[\text{P}]_0}{e^{-\frac{\Delta G_{\text{PM}}}{RT}}[\text{P}]_0 + 1}, 0\right) \searrow \tag{3}$$

$$\text{Specificity}\big([\text{B}]_0, -\Delta G_{\text{BW}}\big) = e^{\frac{\Delta\Delta G_{\text{P}}}{RT}} \times \frac{\left([\text{B}]_0 - c_0\right)e^{-\frac{\Delta G_{\text{BW}}}{RT}} + [\text{P}]_0 e^{-\frac{\Delta G_{\text{PM}}+\Delta\Delta G_{\text{P}}}{RT}} + 1}{\left([\text{B}]_0 - c_0\right)e^{-\frac{\Delta G_{\text{BW}}+\Delta\Delta G_{\text{B}}}{RT}} + [\text{P}]_0 e^{-\frac{\Delta G_{\text{PM}}}{RT}} + 1}$$
$$\in \left[e^{\frac{\Delta\Delta G_{\text{P}}}{RT}} \times \frac{[\text{P}]_0 e^{-\frac{\Delta G_{\text{PM}}+\Delta\Delta G_{\text{P}}}{RT}} + 1}{[\text{P}]_0 e^{-\frac{\Delta G_{\text{PM}}}{RT}} + 1}, e^{\frac{\Delta\Delta G_{\text{P}}}{RT}} \times e^{\frac{\Delta\Delta G_{\text{B}}}{RT}}\right) \nearrow \tag{4}$$

where $\Delta\Delta G_{\text{P}}$ and $\Delta\Delta G_{\text{B}}$ represented the free energy difference of undesired reactions as compared with the corresponding desired reactions, $\Delta G_{\text{BW}}$ and $\Delta G_{\text{BM}}$ were the free energy change of WT + Blocker → BWT and MT + Blocker → BMT, $R$ was the ideal gas constant and $T$ was the temperature in Kelvin. Shown in Eqs. (3) and (4), the sensitivity was monotonically decreasing with $[\text{B}]_0$ and $-\Delta G_{\text{BW}}$ ($\searrow$ meant that the function was monotonically decreasing with its variables), while the specificity was monotonically increasing with $[\text{B}]_0$ and $-\Delta G_{\text{BW}}$ ($\nearrow$ meant that the function was monotonically increasing with its variables). For more visual display, we took a designed probe (Probe-1) as a model, and calculated the involved thermodynamic parameters.

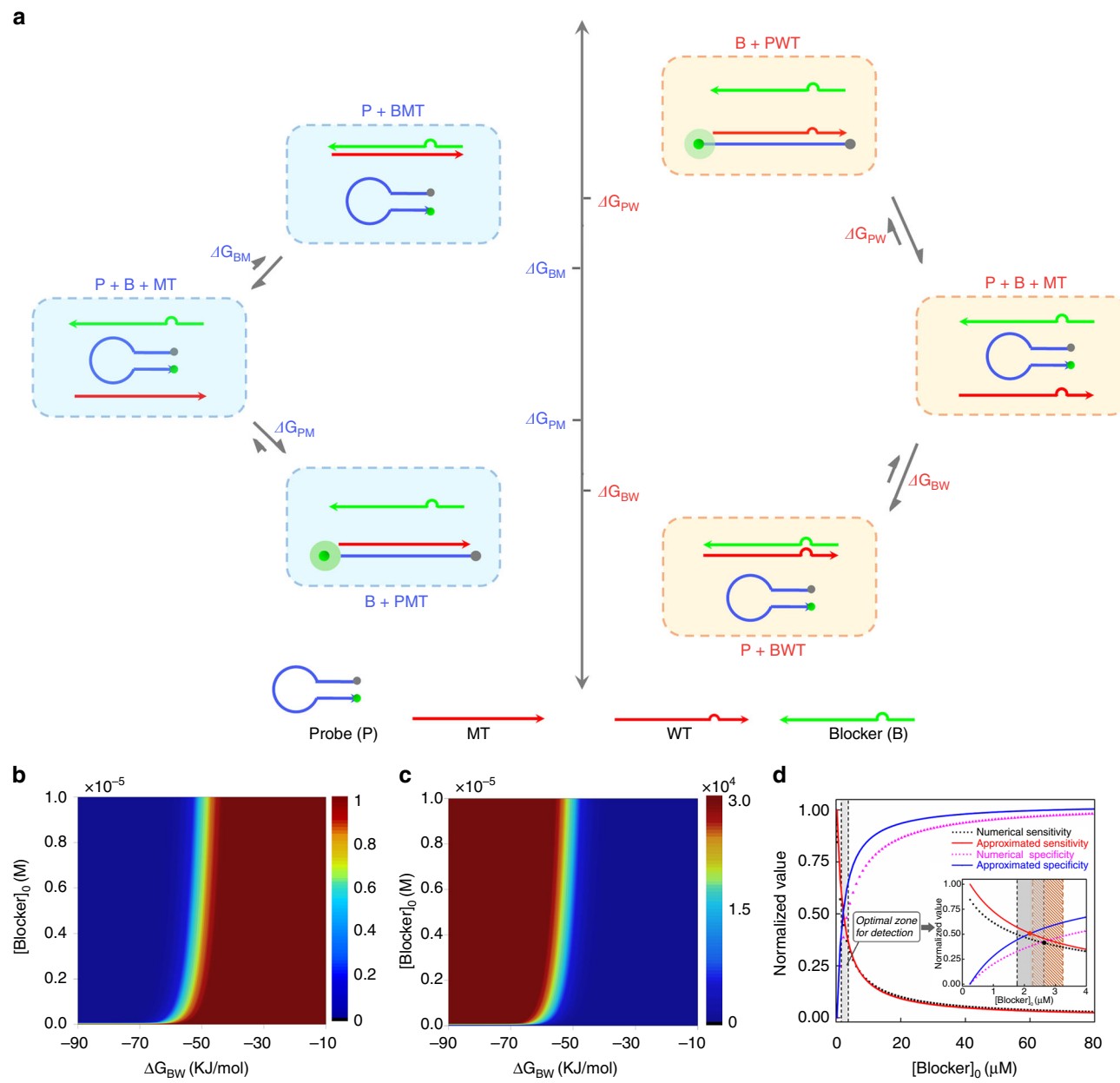

**Fig. 1** Thermodynamic modelling of the Standard probe/standard blocker system. **a** The reaction pathways and the levels of associated free energy changes. Mutanty-type target and Wild-type target were denoted as MT and WT, respectively. **b** The calculated sensitivity of the system over $[Blocker]_0$ (also denoted as $[B]_0$) and $\Delta G_{BW}$. **c** The calculated specificity of the system over $[Blocker]_0$ and $\Delta G_{BW}$. **d** Two-dimensional interception curves of sensitivity and specificity by setting $\Delta G_{BW}$ at a fixed value of $-54596\,\mathrm{J\,mol^{-1}}$

By introducing the parameters into Eqs. (3) and (4), we could draw the heat maps of sensitivity and specificity over $[B]_0$ and $-\Delta G_{BW}$ (Fig. 1b, c). The modelling results clearly showed the inverse correlation of sensitivity and specificity. Fixing the sequence of blocker strand and thereby setting its $-\Delta G_{BW}$ at the corresponding value, we could obtain the two-dimensional interception curves of sensitivity and specificity. Shown in Fig. 1d, the two inversely correlated curves had a balanced point where the sensitivity and specificity could trade off for a comprehensive detection performance. Typically, researchers expected that the sensitivity and specificity of a detection system were both relatively high. Therefore, we defined an optimal function zone, within which the system's sensitivity and specificity were both higher than 80% of the system's sensitivity and specificity at the balanced point. Marked in grey in Fig. 1d, the optimal function

zone of the conventional probe/blocker composition systems was very narrow. Researchers had to make great efforts to optimize the structures and concentrations of blocker strand to reach to the optimal zone, which greatly restrained the application range.

It was worth noting that we also introduced the parameters into the equations without taking any approximations and obtained a series of accurate solutions for plotting the interception curve (scatter curve in Fig. 1d). We could see that as a whole, the approximated curve was very close to the accurate scatter curve especially when $[B]_0$ was high, demonstrating the rationality of the adopted approximations. However, the trouble was that the main deviation between the approximated curve and the accurate curve lied right in the system's optimal detection zone. More terribly, the sensitivity and specificity changed harshly around the balanced point. Therefore, shown in the inset of

Fig. 1d, such deviations rendered the optimal detection zone predicted by approximated curves considerably deflected from that predicted by accurate curves. Moreover, the probe and the blocker could be double-stranded[38–42], and the reactions between targeting strand and probe and blocker were via toehold mediated strand displacement (strand displacement probe/strand displacement blocker composition system). Modelling results in Supplementary Note 2 and Supplementary Fig. 1 showed that such system also possessed inverse correlation between sensitivity and specificity, and the deflection of approximated curves was even larger.

Overall, the conventional probe/blocker composition systems had an intrinsic inverse correlation between the sensitivity and specificity, and the system's performance within the optimal detection zone could not be accurately predicted by thermodynamic models, making the experimental optimization inevitable and cumbersome, especially when multiplexed or high-throughput analysis were needed.

**Strand displacement probe/standard blocker composition system.** To address the above problem, we carefully inspected the thermodynamic model and found that the key reason for the monotonic decreasing of sensitivity was that the blocker strand was completely a negative factor to the system's sensitivity as it could not provide any free energy drive for the formation of PMT at all. Herein, we tried to optimize the probe's structure so that the blocker strand, aside from capturing MT, could in a way provide some free energy drive for the formation of PMT. We then came up with a novel system: strand displacement probe/standard blocker composition system (Fig. 2a).

Define the sensitivity of a toehold probe alone towards targeting strand as $F(x, [PS]_0)$, in which the variable $x$ was the equilibrium constant and the $[PS]_0$ was the initial concentration of the probe. Then (see Supplementary Note 3 for details),

$$F(x, [PS]_0) =$$
$$\frac{([PS]_0 + c_0) \cdot x + [S]_0 - \sqrt{([PS]_0 - c_0)^2 \cdot x^2 + (4 \cdot [PS]_0 c_0 + 2 \cdot [PS]_0 [S]_0 + 2 \cdot [S]_0 c_0) \cdot x + [S]_0^2}}{2 c_0 (x - 1)}$$
(5)

where, $[S]_0$ was the initial concentration of free S strand. Also, we defined,

$$\mu = K_{BW}[B] = e^{-\frac{\Delta G_{BW}}{RT}}[B]$$
(6)

$$\mu_{BM} = K_{BM}[B] = e^{-\frac{\Delta\Delta G_B}{RT}}\mu$$
(7)

When the approximation of $[B] = [B]_0 - c_0$ was adopted,

$$\mu = e^{-\frac{\Delta G_{BW}}{RT}}([B]_0 - c_0)$$

In our experiments, the $[PS]_0$ was constant when tuning the blocker sequence and concentration, so the binary function of $F(x, [PS]_0)$ could be simplified as a unary function of $F(x)$. Then

(see Supplementary Note 3 for details),

$$\text{Sensitivity}([B]_0, -\Delta G_{BW}) = F\left(\frac{\frac{K_{PM}K_{BS}}{K_{BM}}\mu_{BM} + K_{PM}}{\mu_{BM} + 1}\right)$$
$$\in \left[F\left(e^{-\frac{\Delta G_{PM}}{RT}}\right), F\left(e^{-\frac{\Delta G_{PM} - \Delta G_{BW} - \Delta\Delta G_B + \Delta G_{BS}}{RT}}\right)\right) \searrow$$
(8)

$$\text{Specificity}([B]_0, -\Delta G_{BW}) = \frac{F\left(\frac{\frac{K_{PM}K_{BS}}{K_{BM}}\mu_{BM} + K_{PM}}{\mu_{BM} + 1}\right)}{F\left(\frac{\frac{K_{PW}K_{BS}}{K_{BW}}\mu_{BW} + K_{PW}}{\mu_{BW} + 1}\right)}$$
$$\in \left[\frac{F\left(e^{-\frac{\Delta G_{PM}}{RT}}\right)}{F\left(e^{-\frac{\Delta G_{PM} + \Delta\Delta G_P}{RT}}\right)}, \frac{F\left(e^{-\frac{\Delta G_{PM} - \Delta G_{BW} - \Delta\Delta G_B + \Delta G_{BS}}{RT}}\right)}{F\left(e^{-\frac{\Delta G_{PM} + \Delta\Delta G_P - \Delta G_{BW} + \Delta G_{BS}}{RT}}\right)}\right)$$
(9)

It was worth noting that the specificity was not always monotonically increasing over $[B]_0$ and $-\Delta G_{BW}$ under all occasions. For certain thermodynamic parameters, the specificity curve could present small fluctuations when varying $[B]_0$ and $-\Delta G_{BW}$. But even then, the amplitude of the fluctuations was so tiny that as a whole, the specificity seemed to be increasing over $[B]_0$ and $-\Delta G_{BW}$. Equations (8) and (9) showed that although the sensitivity and specificity herein were still inversely correlated, the sensitivity, unlike the conventional probe/blocker composition system, no longer decreased to 0 when $[B]_0$ and $-\Delta G_{BW}$ were very large. This was a significant advance as we could simply increase the length and concentration of blocker strand to enhance the system's specificity without the concern of too much loss of sensitivity. The optimal zone for detection was very wide, and thereby the optimization process could be simplified. However, the experimental results were not in accordance with the modelling results. Taking a designed probe (Probe-3) as a model, we investigated the variation of the system's sensitivity over the length and concentration of blocker strand. We would like to note that the thermodynamic parameters of probe-3 were appropriate and the specificity was rigorously increasing over $[B]_0$ and $-\Delta G_{BW}$. Shown in Fig. 2e and Supplementary Fig. 5, when the concentration and length of the blocker increased to a large value, the increase of fluorescence intensity, which directly represented the system's sensitivity, became very little (close to 0), which was largely deviated from the calculated sensitivity of ~0.5. We attributed this discrepancy to the system's kinetics that within a typical reaction time of <3 h, the reaction system was far from reaching to thermodynamic equilibrium. Typically, in real detection, the MT or WT would firstly react with blocker. Then, the probe was introduced for subsequent reaction. Take MT for instance, the first reaction step of MT + Blocker ⇌ BMT (reaction 1 and 2) could quickly reached to a state that was very close to the equilibrium state (precisely, it would take very long time to reach to the exact equilibrium state as the reverse reaction was very slow). Then, the probe was added and the aforementioned thermodynamic model expected reactions 1–6 to adequately proceed and reach to global equilibrium within the detection period. However, the kinetic constant of reaction 2 was so small (~$10^{-35}$) that it could be regarded as quasi-non-reactable especially when $[B]_0$ and $-\Delta G_{BW}$ were very large[32,38,39]. In the thermodynamic model, reaction 2 was the only pathway for BMT to transform to other species and this path was hindered by the ultra-slow kinetics. Therefore, the whole reaction system seemed to be trapped in BMT + PS. Under such conditions, we could reasonably assume that the equilibriums of reactions 1–2 and reactions 3–6 were independent and would not interfere with each other. We then calculated the sensitivity and specificity over $[B]_0$ and $-\Delta G_{BW}$ under such assumptions (see Supplementary

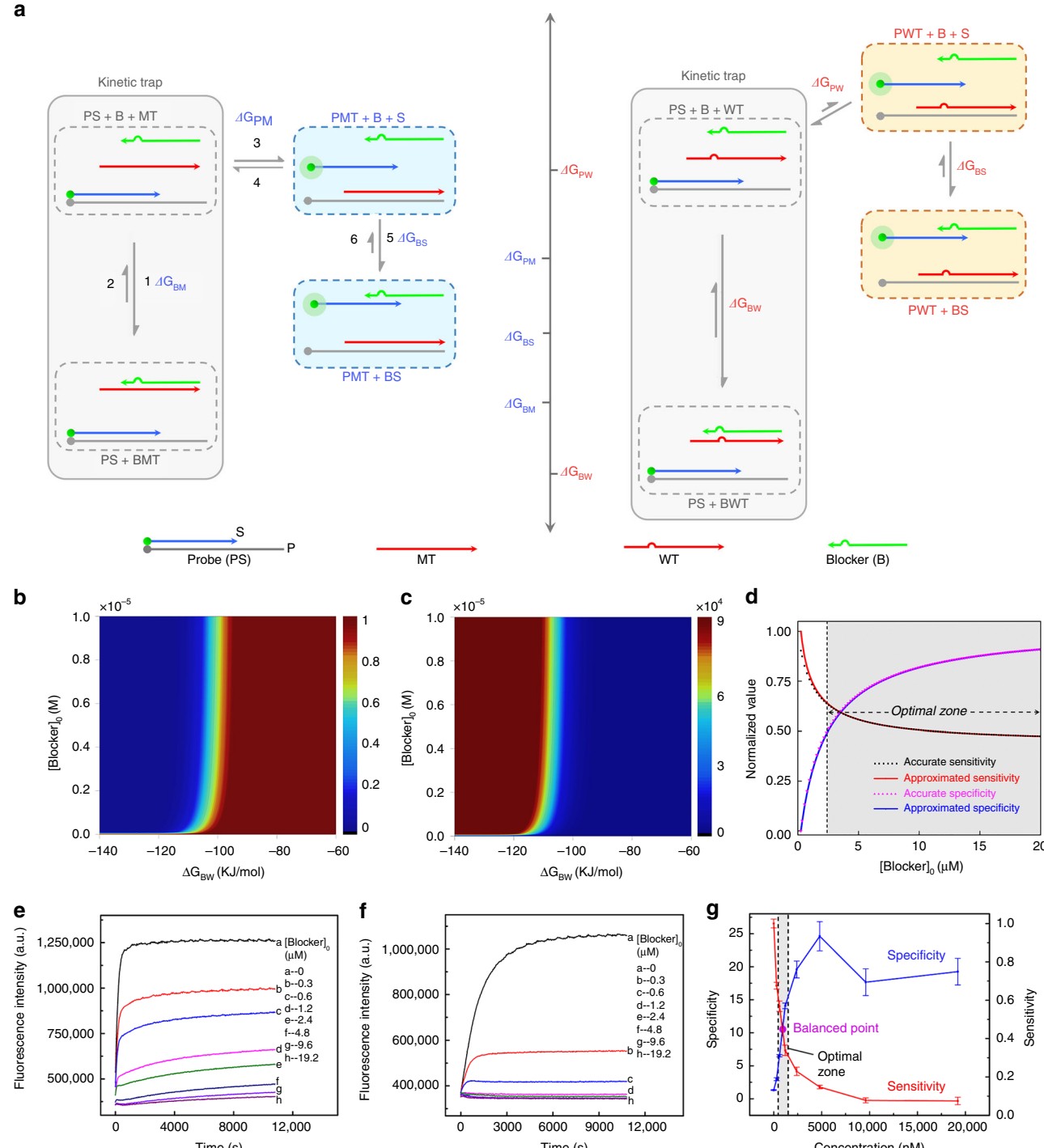

**Fig. 2** Modelling and verification of the Strand displacement probe/standard blocker system. **a** The reaction pathways and the levels of associated free energy changes. **b** Calculated sensitivity of the system over [Blocker]$_0$ and $\Delta G_{BW}$. **c** Calculated specificity of the system over [Blocker]$_0$ and $\Delta G_{BW}$. **d** Two-dimensional interception curves of sensitivity and specificity extracted from (**b**) and (**c**). The value of $\Delta G_{BW}$ was fixed at $-101403$ J mol$^{-1}$. **e–f** The fluorescence intensity of system towards MT (**e**) and WT (**f**) with the concentration of blocker ranging from 0 to 19.2 μM. The length of blocker was fixed at 18-nt. **g** The sensitivity and specificity of the system over [Blocker]$_0$. Error bars are s.d. of three repeated experiments. Source data are provided as a Source Date file

Figs. 2 and 3). Modelling results showed that as [B]$_0$ and $-\Delta G_{BW}$ increased, the sensitivity decreased to around 0 whereas the specificity increased to the upper limit, in accordance with the experimental results.

Based on the above kinetic analysis, we could see that although the strand displacement probe/standard blocker composition system had provided a good scenario for alleviating the inverse correlation in thermodynamics, the kinetic trap almost completely blocked it,

making the design invalid and the calculation highly inaccurate. Therefore, addressing the key problem required comprehensive consideration of both the thermodynamics and kinetics.

If we carefully inspected reaction pathways in Fig. 2, we could find that the best way to break up the kinetic trap was to create a new reaction pathway for BMT + PS to transform into PMT + BS. In the current design, there was no toehold domain to initiate the reaction. Quite naturally, we came up with the idea of moving

the blocker sequence to spare out a toehold domain, and thereby MT could escape from the BMT to PMT through 4-way strand exchange. Moreover, the value range of the system's sensitivity (Eq. (8)) provided important guidance that if we move the blocker sequence to an appropriate domain, $\Delta G_{BM}$ ($\Delta G_{BW}$ + $\Delta\Delta G_B$) could be identical to $\Delta G_{BS}$ and the system's sensitivity could be constant.

**Four-way Strand Exchange LEd Competitive DNA Testing system.** Guided by the thermodynamics and kinetics of the system in Fig. 2, we moved the blocker sequence to be restrained within the branch migration domain. Thus, the sequences of the hybridization domains of BPs and BMT were exactly the same, which guaranteed $\Delta G_{BM} = \Delta G_{BPs}$. As for the kinetics, restriction of the blocker within the branch migration domain spared out a single-stranded domain in BMT for initiating the 4-way strand exchange process. The reaction pathways and the free energy change landscape were illustrated in Fig. 3a.

Using the mass-action equilibria, we could obtain the expression of sensitivity (see Supplementary Note 4 for details), and it was determined only by the parameters of probes, which remained constant when blocker sequence and concentration were to be optimized. As for the specificity, it would monotonically increase to its theoretical upper limit as the $[B]_0$ and $-\Delta G_{BW}$ increased. Overall, the established thermodynamic model indicated that the 4-way SELECT system did not suffer from the inverse correlation between sensitivity and specificity.

$$\text{Sensitivity}\big([B]_0, -\Delta G_{BW}\big) = F\left(\frac{\mu_{BM}K_{PBM} + K_{PM}}{\mu_{BM} + 1}\right)$$
$$= F\left(e^{-\frac{\Delta G_{PM}}{RT}}\right) \equiv \text{constant} \tag{10}$$

$$\text{Specificity}\big([B]_0, -\Delta G_{BW}\big) = \frac{F(K_{PM})}{F\left(\frac{\mu K_{PBW} + K_{PW}}{\mu + 1}\right)}$$
$$= \frac{F\left(e^{-\frac{\Delta G_{PM}}{RT}}\right)}{F\left(\frac{e^{\frac{\Delta G_{PM} + \Delta\Delta G_P + \Delta\Delta G_B}{RT}}\left([B]_0 - c_0\right) + e^{-\frac{\Delta G_{PM} + \Delta\Delta G_P}{RT}}}{e^{-\frac{\Delta G_{BW}}{RT}}\left([B]_0 - c_0\right) + 1}\right)}$$
$$\in \left[\frac{F\left(e^{-\frac{\Delta G_{PM}}{RT}}\right)}{F\left(e^{-\frac{\Delta G_{PM} + \Delta\Delta G_P}{RT}}\right)}, \frac{F\left(e^{-\frac{\Delta G_{PM}}{RT}}\right)}{F\left(e^{-\frac{\Delta G_{PM} + \Delta\Delta G_P + \Delta\Delta G_B}{RT}}\right)}\right] \nearrow \tag{11}$$

It was worth noting that the derivation of Eq. (10) did not adopt any approximation at all. For specificity, we could prove it to be monotonically increasing with the equilibrium concentration of blocker ($[B]$) and $-\Delta G_{BW}$ (see Supplementary Note 4 for details), so the value range shown in Eq. (11) were accurate results with no approximation. But we could not judge the monotonicity of it over $[B]_0$ due to the extreme complexity of the analytic expressions. Therefore, for the derivation of the analytic expression and monotonicity in equation (20), we adopted the approximation of $[B] = [B]_0 - c_0$. We then took a designed probe (Probe-4) as a model and obtained the heat maps and intersection curves of specificity and sensitivity (Fig. 3b–d). The results demonstrated the 4-way SELECT system could break up the inverse correlation between sensitivity and specificity. And the wide overlap between the curves with and without approximation demonstrated the rationality of the adopted approximation.

Looking over Eqs. (10) and (11), we could elucidate the essence of the 4-way SELECT system: The influence of blocker on the hybridization thermodynamics seemed to be integrated into the hybridization thermodynamics of the toehold probe, and the 4-

way SELECT system as a whole could be equivalent to a hypothetic toehold probe alone with an equilibrium constant of $(\mu_T K_{PT} + K_{PBT})/(\mu_T + 1)$, where $T$ was the targeting strand, $K_{PT}$ and $K_{PBT}$ were the equilibrium constants of PS + T ⇌ PT + S and PS + BT ⇌ PT + BS, respectively, and $\mu_T$ was $K_{BT}[B]$. Increasing the concentration and length of the blocker was actually changing the intrinsic equilibrium constant of the hypothetic toehold probe from the upper limit ($K_{PM1}$ or $K_{PW1}$) to the lower limit ($K_{PBM}$ or $K_{PBW}$). Overall, the whole 4-way SELECT system was equivalent to a single toehold probe with its equilibrium constants towards MT and WT regulated by blocker, and this equivalence brought out significant advantages of the SELECT system over conventional probe/blocker composition systems (see Supplementary Note 5).

The above thermodynamic modelling results were satisfying. However, as we have discussed, the intrinsic inverse correlation also derived from the kinetic traps. When the blocker strand was relatively long, reactions 2 and 6 would be very slow and quasi-non-reactable, thus making the whole system trapped in a vast majority of BMT + PS. Owing to the spared ssDNA domain in BMT, the kinetic trap could be broken through a new reaction pathway between BMT + PS and PMT + BS: the 4-way strand exchange (reactions 7 and 8). When $[B]_0$ and $-\Delta G_{BW}$ were small, the whole system could reach to global equilibrium through reactions 1–8. However, if $[B]_0$ and $-\Delta G_{BW}$ were large, which was the common situation in real applications, the system was not able to reach to the global equilibrium due to the ultra-slow kinetics of reactions 2 and 6. Consequently, the vast majority of the species were trapped within the localized equilibrium of reactions 7–8 within several hours. Fortunately, the free energies of BMT + PS and PMT + BS were significantly lower than other species. So, at the final global equilibrium state, the vast majority of the species in the system were exactly BMT, PS, PMT and BS. Thus, the localized equilibrium of reactions 7–8 was very close to the global equilibrium of reactions 1–8. Therefore, the established thermodynamic model, although based on global equilibrium, could describe the system at the localized equilibrium state with little deviations.

The vital point for the 4-way SELECT system was that the 4-way strand exchange (reactions 7 and 8) could happen within a typical detection duration of several hours. To verify it, we synthesized MT-1, WT-1, probe-1 and corresponding blocker strands with the length ranging from 12 to 18 (denoted as blocker-1-12 to blocker-1-18). Firstly, we fixed the concentration of the blocker strand, probe-1 and MT-1/WT-1 at 2 μM, 1 μM and 250 nM, respectively, and investigated the influence of the length of blocker strand on the system's sensitivity. Shown in Fig. 3h, as the length of blocker strand increased, the increase rate of the fluorescence intensity became smaller, but the fluorescence intensity plateau, which represented the sensitivity, were nearly unchanged. We then fixed the length of the blocker strand at 18 and changed its concentration from 250 nM to 5 μM. We observed similar phenomenon that the sensitivity remained unchanged (Fig. 3e). These results demonstrated that the system's sensitivity was a constant regardless of $[B]_0$ and $-\Delta G_{BW}$, in accordance with the thermodynamic model. This accordance also confirmed that the 4-way strand exchange could finish within a short period of time. To go a step further, we performed the above experiments on WT-1. Results depicted in Fig. 3f and i showed that the fluorescence intensity plateau decreased as the $[B]_0$ and $-\Delta G_{BW}$ increased. We then could draw the curves of the system's sensitivity and specificity over $[B]_0$ and $-\Delta G_{BW}$ (Fig. 3g and j). Compared with the calculated curves (Fig. 3d), we could see that the variation trends (the shapes and inflection points of the curves) of sensitivity and specificity in Fig. 3g were quite similar, demonstrating the accordance between modelling

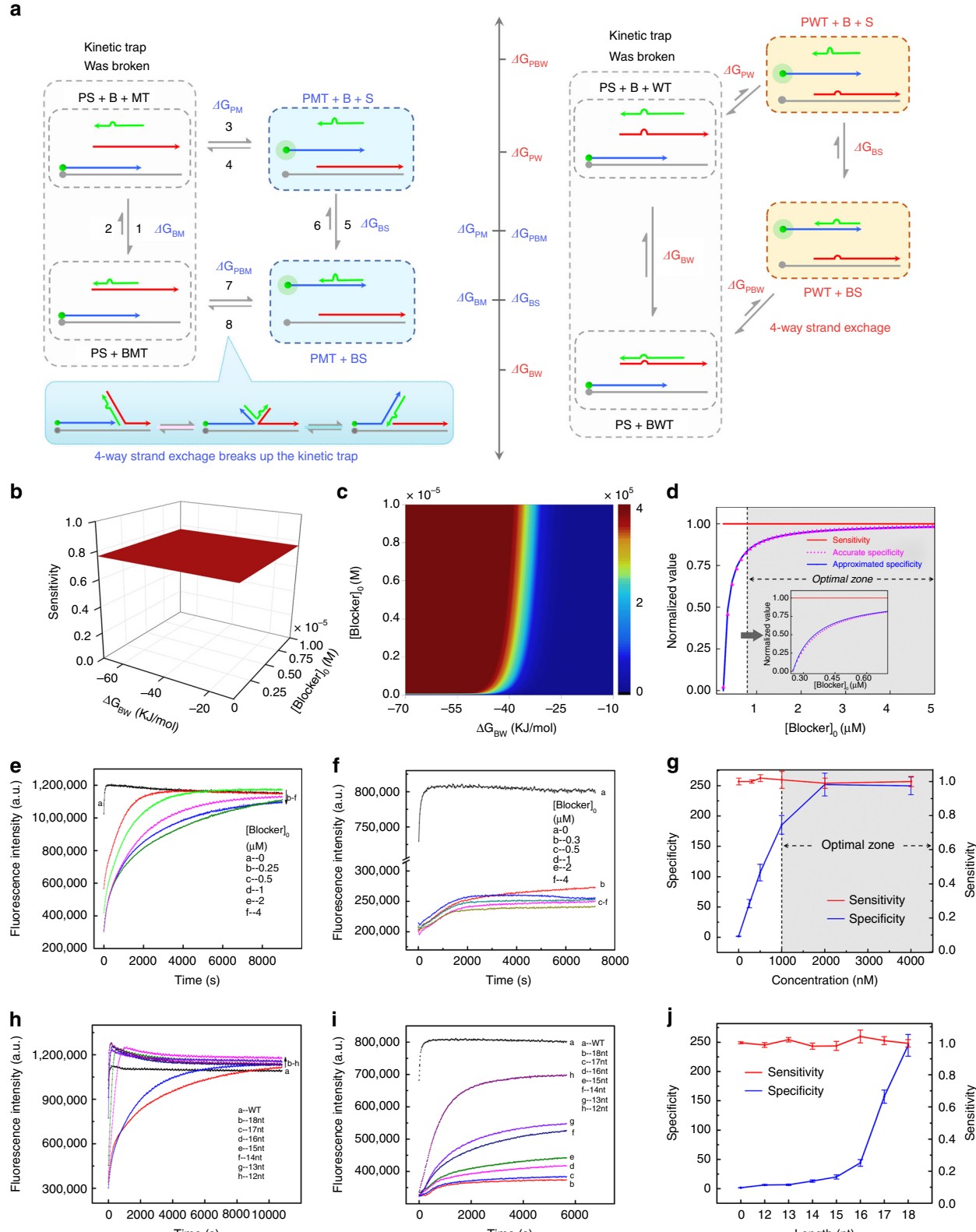

**Fig. 3** Modelling and verification of the dissociative 4-way SELECT system. **a** The reaction pathways and the levels of associated free energy changes. **b–c** Calculated sensitivity (**b**) and specificity (**c**) of the system over [Blocker]$_0$ and $\Delta G_{BW}$. **d** Two-dimensional interception curves of sensitivity and specificity extracted from (**b**) and (**c**). The value of $\Delta G_{BW}$ was fixed at $-71421$ J mol$^{-1}$. **e, f** The fluorescence intensity of the system towards MT (**e**) and WT (**f**) with the concentration of blocker ranging from 0 to 4 μM. The length of blocker was fixed at 18-nt. **g** The sensitivity and specificity of the system over [Blocker]$_0$. **h–i** The fluorescence intensity of the system towards MT (**h**) and WT (**i**) with the length of blocker ranging from 12 nt to 18 nt. The concentration of blocker was fixed at 4 μM. **j** The sensitivity and specificity of the system over blocker length. Error bars are s.d. of three repeated experiments. Source data are provided as a Source Date file

and experimental results. We noted that the upper limit of the experimental specificity was considerably lower than the calculated specificity, and we owed this discrepancy to the limitation of the fluorescence signal intensity that could be detected by the instrument, the inherent noise of the instrument and the inevitable errors alongside the whole detection process. These factors prevented us from detecting very slight fluorescent signals produced by traces of PWT and set an upper limit of specificity that could be accurately measured. In consideration of the signal window and the standard deviations observed in the experiments, we could estimate the upper limit to be around 250, and specificities calculated to be higher than 250 was experimentally inaccurate. Based on the above discussion, we have theoretically and experimentally demonstrated that the 4-way SELECT system could break the inverse correlation between sensitivity and specificity.

The probe of the above composition system was in the dissociative mode, in which the S strand would dissociate off the probe after branch migration. Another commonly used mode was the non-dissociative mode, in which the docking domain and dissociation domain were so long that after branch migration, the S strand stayed within the probe. The reaction pathways and free energy landscapes were depicted in Supplementary Fig. 4. We then could obtain the expressions of sensitivity and specificity and their monotonicity (see Supplementary Note 6 for details),

$$\text{Sensitivity}\big([\text{B}]_0, -\Delta G_{\text{BW}}\big) \equiv 0.5 \qquad (12)$$

$$\text{Specificity}\big([\text{B}]_0, -\Delta G_{\text{BW}}\big) = \frac{0.5}{G\left(\frac{\mu K_{\text{PBW}} + K_{\text{PW}}}{\mu + 1}\right)}$$

$$\in \left[\frac{1}{2G\left(e^{-\frac{\Delta G_{\text{PM}} + \Delta \Delta G_{\text{P}}}{RT}}\right)}, \frac{1}{2G\left(e^{-\frac{\Delta G_{\text{PM}} + \Delta \Delta G_{\text{P}} + \Delta \Delta G_{\text{B}}}{RT}}\right)}\right) \nearrow$$

$$(13)$$

where, $G(x)$ was defined as $x/(x+1)$. $G(x)$ was actually the sensitivity of a non-dissociative strand displacement probe alone towards targeting strands, in which the variable $x$ was the equilibrium constant. Taking one specific reaction system as a model, we drew the heat maps and interception curves of the system's sensitivity and specificity over $[\text{B}]_0$ and $-\Delta G_{\text{BW}}$. We then performed experiments and results shown in Supplementary Fig. 6 were in accordance with the theoretical calculation.

**Fluorescent signal at low variation frequencies**. In clinical applications, mutations could present low frequency in many cases[43,44], i.e. the percentage of MT in the mixture of MT/WT was low. Especially in the tumour tissue, only a small fraction of cells within it carried the targeting genetic variation (mutation), so the genetic DNA extracted from the tumour tissue present low variation frequency. To assess the ability of the 4-way SELECT system in such applications, we defined a novel parameter: the relative increase of fluorescence intensity at low variation frequencies (abbreviated as IF). Illustrated in Fig. 4a, the IF values were calculated from:

$$\text{IF} = \frac{F_a}{F_b} \times \frac{1}{\text{VAF}} \qquad (14)$$

where VAF was the frequency of the targeting mutation. In real detection, the signal of the unknown sample was compared with that of the negative control (pure WT), and the larger difference between their signals, the better detection performance we could achieve. Herein, the numerator in the definition of IF ($F_a$) was defined as the difference of fluorescence intensity between the negative control and the sample with a mutation frequency at VAF. $F_a$ could actually be regarded as the signal of the detection.

The denominator in the definition of IF ($F_b$) was defined as the difference of fluorescence intensity between the negative control and the blank control. Thus, $F_b$ could be regarded as the background of the detection. Taken together, IF represented the signal-to-background ratio of a detection system towards samples with mutation frequencies of VAF. It was worth noting that the hybridization efficiencies of probes towards different concentrations of targeting strands were not rigorously proportional, especially when the concentration varied by two or more orders. Therefore, the IF value would change along with the mutation frequencies (different mutation frequencies meant different concentrations of MT when the total amount of DNA was fixed), which was actually beneficial for accurately assessing the system's performance in different occasions. Whereas, the parameter specificity was constant and not suitable for low VAF.

We then used the 4-way SELECT system to detect MT-1 at frequencies of 1%. Shown in Fig. 4a, the IF values were calculated to be 688. Compared with specificity, the IF was significantly higher, which reveal that IF values could better reflect the system's ability in detecting low-frequency mutations.

**Generality towards different types of mismatches**. The generality towards different types of mismatches determined the application range of the 4-way SELECT system. We then tested the system on six types of mismatches. Figure 4b showed the system's uniformly excellent performance towards all tested mismatches (see Supplementary Figs. 7–12 for fluorescent kinetic traces). It was worth noting that the sequence design for all involved strands and the reaction conditions for all experimental groups were completely the same, which firmly demonstrated the outstanding convenience of our system.

**Post-PCR detection of low-abundance point mutation**. As was mentioned above, detecting low-frequency mutations was vital for cancer diagnosis. Taking *KRAS* G12S and G13D as modelling mutations, we used the 4-way SELECT system to detect synthesized 40-nt MT strand with different frequencies. Supplementary Fig. 13 showed that both the dissociative and non-dissociative 4-way SELECT system could detect 0.1% *KRAS* G13D and 0.1% *KRAS* G12S mutation, in accordance with the IF values. In real detection, the 4-way SELECT system was performed on PCR amplicons. We then needed to assess the compatibility of our system with PCR. Shown in Fig. 4c, a serial of mixed samples with different mutation frequencies were prepared and PCR amplified. The PCR products were treated with exonuclease I and Lambda exonuclease to yield single-stranded DNA for subsequent analysis. Figure 4d, e plotted the fluorescence intensity responses of the dissociative and non-dissociative 4-way SELECT system towards targeting mutations with different frequencies. We could clearly see that the detection limits were 0.1% for both systems, in agreement with the detection limits on synthesized targeting strands. These results further verified the ultra-high discrimination ability of our method and firmly demonstrated the feasibility of post-PCR detection.

**Multiplexed identification of SNVs**. The most significant advantage of our proposed composition system was the great convenience in sequence design and condition optimization, which was vital for multiplexed or high-throughput analysis. To demonstrate this, we utilized it to identify 16 kinds of mutations with clinical significance. It was important to point out that for all the 16 kinds of mutations, we set the principles of sequence design and the reaction conditions to be exactly the same, thus making it optimization-free multiplexed detection. Moreover, to significantly lower down the expenses, we designed the Y-shaped

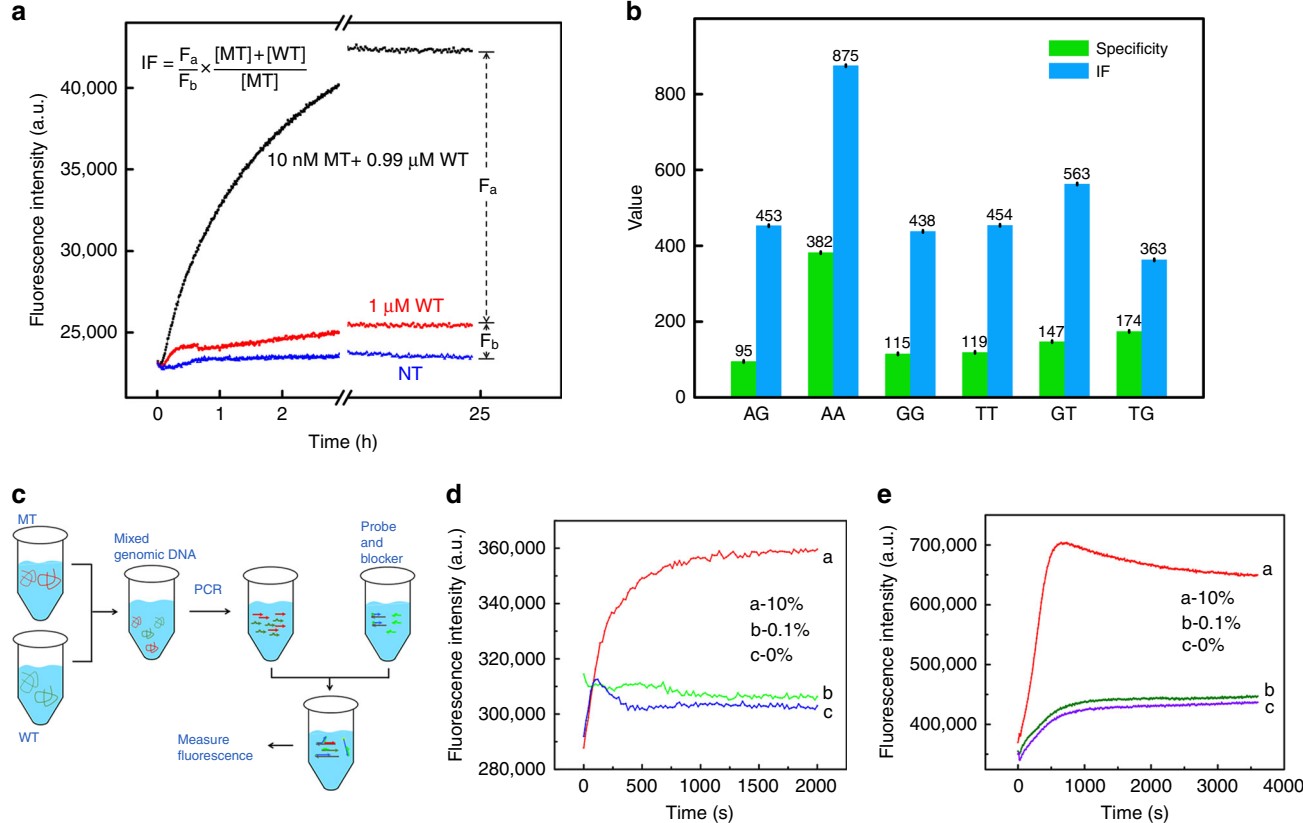

**Fig. 4** Functionality and clinical practicability of the 4-way SELECT system. **a** Definition of the relative increase of fluorescence intensity at low variation frequencies (abbreviated as IF). **b** The specificity and IF values of dissociative 4-way SELECT system towards six different types of mismatches. **c** The workflow of post-PCR detection of low-abundance point mutation. **d**, **e** Detection of low-abundance point mutations using the dissociative 4-way SELECT system (**d**) and the non-dissociative 4-way SELECT system (**e**). Source data are provided as a Source Date file

probe in which the fluorophore labelled strand and quencher labelled strand could be shared among different probes targeting different mutations[32]. Experimental results in Fig. 5a showed that the uniformed 16-plex 4-way SELECT system was able to identify all the tested mutations with high specificities (see Supplementary Fig. 14 for fluorescent kinetic traces). For clinical validation, we utilized our system on genomic DNAs extracted from the blood samples of an ovarian cancer patient and a healthy volunteer. Figure 5b and Supplementary Fig. 13e, f showed that the results provided by the 4-way SELECT system were in accordance with the Sanger sequencing results, demonstrating the practicability of our system in real clinical applications.

## Discussion

We firstly would like to discuss the main approximations we have adopted in the theoretical modelling[45]: (i) We adopted the approximation of $[B] = [B]_0 - c_0$, which, for conventional probe/blocker competitive systems, required that the bonding of blocker to target greatly out-compete that of probe to target ($K_{BT}[B]_0 \gg K_{PT}[P]_0$). However, for the 4-way SELECT system, the assumption only required that bonding of blocker to target is strong ($K_{BT} \gg 0$), and the blocker did not need to out-compete probe because the competition between them was eliminated by the formation of BS. In the experiments concerning the influence of $[B]_0$ on the system's sensitivity and specificity, the $K_{BW}$ were fixed at a very large value, so $[B] = [B]_0 - c_0$ was satisfied under all situations from low $[B]_0$ ($c_0$) to high $[B]_0$ ($+\infty$). For the experiments concerning the influence of $K_{BW}$ on the system's sensitivity and specificity, $[B] = [B]_0 -$

$c_0$ was completely feasible when $K_{BW}$ was large, which assured an accurate prediction of the upper limit of specificity. Though there was deviation in the calculation of specificity when $K_{BW}$ was small, it was not important because we certainly would not apply our system with its specificity at such low values. (ii) All the theoretical calculation assumed that the reactions of MT and WT were independently happening in two individual tubes, in which the equilibrium concentration of blocker strand should be a little bit different between MT and WT. This assumption was completely true for measuring specificities or genotyping SNVs. However, for measuring IF values or the detection of low-abundance point mutations, MT were submerged in the excessive WT, which meant that the reactions of MT and WT were in the same tube and the blocker concentration was the same for both of them. Actually, this deviation could also be eliminated by a large excessive concentration of blocker strand, in which $[B] = [B]_0 - c_0$. Since we used same $[B]_0$ for MT reaction and WT reaction, the equilibrium blocker concentrations would be identical for both of them. Overall, the established thermodynamic model could accurately predict the system's performance in low-abundance mutation detection.

Another vital point we would like to discuss was the kinetics of the 4-way strand displacement happening between BWT and probe (BWT + probe ⇌ PWT + BPs). As was demonstrated by experimental results, the 4-way strand displacement process happening between BMT and probe (BMT + probe ⇌ PMT + BPs) was relatively fast. However, the free energy change of the reaction of BWT + probe ⇌ PWT + BPs was considerably more positive than that of BMT + probe ⇌ PMT + BPs due to the two

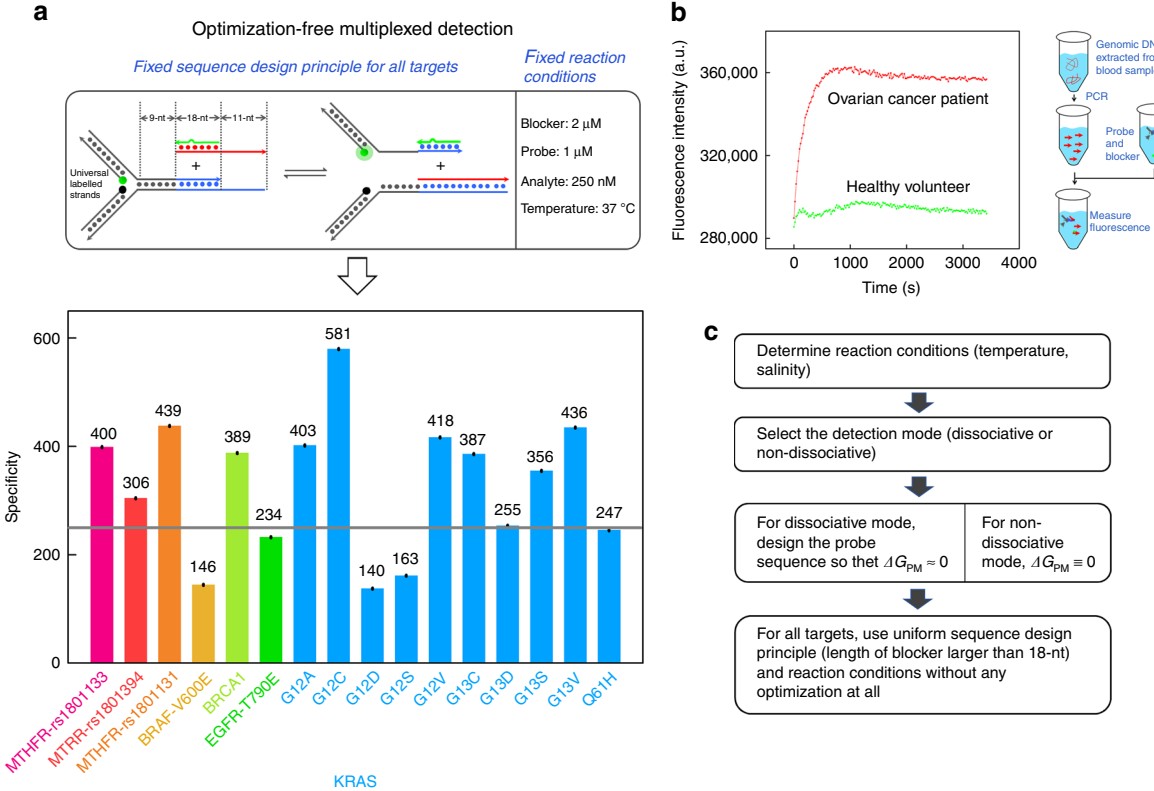

**Fig. 5** Multiplexed detection by the 4-way SELECT system. **a** Optimization-free multiplexed detection of 16 kinds of genetic variations. The fluorophore and quencher of the probe was designed to be universal, and the sequence design principle and reaction conditions were fixed to be exactly the same towards all targets. **b** Post-PCR genotyping using the universal 4-way SELECT system. **c** The workflow for designing a 4-way SELECT system. Source data are provided as a Source Date file

single-base mismatches in PWT and BPs, respectively. Therefore, the forward reaction rate of $BWT + probe \rightleftharpoons PWT + BPs$ must be lower than that of $BMT + probe \rightleftharpoons PMT + BPs$. Possibly, the reaction of WT was not able to reach thermodynamic equilibrium within the detection duration of several hours. So the experimental specificity of the 4-way SELECT system obtained within several hours could be considerably higher than calculated values. It was worth noting that although it brought deviations to the model, such deviations were actually in the favour side of the system's performance. More importantly, this deviation would not change the monotonicity of the system's sensitivity and specificity over $[B]_0$ and $-\Delta G_{BW}$, which was the most significant advance our system provided to the field.

To sum up, we have thoroughly analysed the reaction mechanisms of competitive DNA hybridization, and constructed complete theoretical models for them. Guided by the theoretical calculation, we developed the 4-way SELECT system which was able to break the intrinsic inverse correlation between sensitivity and specificity over blocker sequence and concentration. This intrinsic inverse correlation had long been the bottleneck of the area of probe based nucleic acid analysis. Our proposed probe design approach solved this bottleneck and showed great convenience and robustness in nucleic acid analysis especially for multiplexed or high-throughput mutation detection. Scientifically, our work greatly deepened our understanding about the thermodynamics and kinetics of DNA hybridization, and the constructed theoretical model could provide guidance for researchers to design DNA based assays or tools. Practically, the established 4-way SELECT system served as a very powerful tool for DNA analysis, especially for multiplexed or high-throughput mutation detection.

## Methods

**Theoretical calculation and modelling**. For theoretical calculation and modelling, the free energy changes of each reactions were estimated by NUPACK. The Sequences of Oligonucleotides used for modelling were shown in Supplementary Table 6, and their thermodynamic parameters were shown in Supplementary Table 7. By introducing the parameters into the corresponding equations, the analytic expressions of sensitivity and specificity, the accurate solutions of mass-action equilibria and the monotonicity of the sensitivity and specificity could be calculated in Maple 2019. Further, the three-dimensional surfaces and two-dimensional curves of sensitivity and selectivity could be drawn using Origin 8.0.

**Probe preparation and reaction setups**. To prepare the Probes, we mixed the Q and F strands at 1:1 ratio in 1× ThermoPol reaction buffer (New England Biolabs). The Probe mixtures were then thermally annealed using PCR machine, following a thermal profile of initial heating to 95 °C for 5 min, and subsequent cooling to 20 °C for 75 min. Prepared probe solutions were then stored in 4 °C until use.

**Reaction setup for probe/blocker composition system**. To plate strip, appropriate amount of Probe, Blocker, WT and MT solution were added (see Supplementary Tables 8–13 for detailed sequences). Then the plate strip was immediately put into a microplate reader (Biotek) for fluorescence measurement. The amount of Probe, Blocker, MT, and WT added in different competitive composition systems are listed in Supplementary Tables 1–3.

**Time-based fluorescence acquisition**. Time-based fluorescence data were acquired using Synergy HTX Multi-Mode Microplate Readers. To minimize the effects of machine-to-machine variability, all experiments on one target are performed on the same instrument. Observed specificity was consistent between the two machines, exhibiting <5% deviation.

We set the excitation wavelength at 485 nm and emission wavelength at 528 nm for FAM. The duration of fluorescence acquisition varies. Typically, we set a maximum acquisition duration of 4 h, but terminate the experiment early if it appears that the reaction has roughly reached equilibrium (in order to facilitate more rapid collection of data). In real applications, however, we were primarily aimed at observing the difference between the unknown sample and control, so we

could terminate the detection when the goal has been achieved even if the reaction has not reached equilibrium.

**Post-PCR genotyping and detection of low-abundance mutations**. The clinical samples were obtained from the Wuhan Union hospital under the approval of the ethics committee of Tongji Medical College of Huazhong University of Science and Technology. Our study is compliant with the "Guidance of the Ministry of Science and Technology (MOST)" for the Review and Approval of Human Genetic Resources. The analyte genomic DNA diluted to a concentration of 50 ng μl$^{-1}$, and 1 μL of the diluted sample was added into a PCR tube to achieve a final concentration of 2.5 ng μl$^{-1}$ (20 μL). The detailed reactants for PCR were listed in Supplementary Tables 4 and 5. After PCR, the products were treated with Exonuclease I and Lambda exonuclease to prepare single-stranded analyte for subsequent probe detection. All experiments were conducted in triplets.

**Reporting summary**. Further information on research design is available in the Nature Research Reporting Summary linked to this article.

## Data availability

All the data supporting the findings of this study are available within the paper and the supplementary files. The source data underlying Figs. 2–5 and Supplementary Figs. 5–14 are provided as a Source Data file. All other relevant data are available from the authors upon reasonable request.

## Code availability

Theoretical modelling was performed on Maple 2019, and the code for reproducing the modelling results could be found via the link https://doi.org/10.6084/m9.figshare.9848438.v1

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

## Acknowledgements

This work was financially supported by the National Science Foundation of China (No. 81871732 and 21705053), the Key Technology Innovation Program of Hubei Province (2019ACA138), the Natural Science Foundation of Hubei Province (No. 2017CFB117), Hubei Province health and family planning scientific research project (No. J2017Q017), and Wuhan Youth Science and Technology Plan (2017050304010293).

## Author contributions

X.C., N.L. and L.L. contributed equally to this work. They performed the theoretical analysis, designed the experiments, conducted most of the experiments, analysed the data. W.C., N.C., M.L., J.X. and X.Z. conducted some of the experiments and participated

in the discussion of the work. X.X. and M.Z. conceived the project and wrote the paper. H.W. provided clinical samples and guidance. X.X. supervised the whole work.

## Competing interests

The authors declare no competing interests.
