## [Peer Review File · Nature Communications]

Reviewers' Comments:

Reviewer #1:

Remarks to the Author:

The authors present a new design of nucleic acid detection systems that allows for high sensitivity and selectivity without requiring fine-tuning of the sequences and concentrations of components. The basic concept is to have the binding of the target to a probe of \sim neutral ΔG relative to the binding of the same probe to a blocker strand, which ensures that there is always a good sensitivity and discrimination relative to a mismatched strand. The idea is an interesting one, and I believe that their proposal is novel and functional (although I am not primarily an experimentalist, and so I would defer to input in that regard). In principle, such work could be published in Nature Communications.

However there are two major problems with the paper as it stands, both of which would require extensive work to make the manuscript suitable. Specifically, the theoretical modelling used to motivate and analyse the experiments is poorly-justified and faulty in places. Secondly, the manuscript is not well written or structured, making it very hard to follow the argument.

1. Theoretical modelling.

- The authors describe their system through the "state" of a single target strand, with its different bonding configurations assigned free energies. Formally, this works ok if all "states" correspond to distinct bonding configurations of a single copy of that strand with other strands that are in enormous excess. However, that is not the case in the authors' work:

(a) Frequently, the blocker strands are not in excess.

(b) More worryingly, the distinct "states" of the model for the later designs (Fig. 2, Fig. 3) are not distinguished by the confirmation of the target strand alone: PM2 and PM1 are differentiated by the state of another strand.

The authors should instead go back to the fundamental mass-action equilibria, eg:

$$[AB]/[A][B] = \exp(-\Delta G^0_{AB}/RT),$$

and derive the steady states properly. I have been through the mathematics myself and I believe that the conclusions they draw about the expected equilibria of the systems illustrated in figures 2 and 3 are wrong. In particular, the tendency of the sensitivity to 1/2 when the conditions of eq. 25,26 are satisfied is wrong. In the limit of a large excess of reporter and blocker, the sensitivity should tend towards 1, not 1/2. Note that, in fact, this is what happens in the graphs in figure 3 - a discrepancy on which the authors do not comment. I suspect the results derived for the specificity are also incorrect.

Indeed, I believe to some extent that the authors "miss the point" about how their system works. The target strands will tend to bind to the reporters even with an excess of blockers, and ~ 0 ΔG difference, because there is an entropic drive to transfer the blue strands from the reporter complex to a complex with the blocker. This is what allows the specificity to approach 1, but it will also have consequences for the selectivity.

2. The writing of the paper.

The paper needs to be thoroughly edited for use of English, but that is not a major problem at this stage. The big issue is that the ideas are not well-presented in a way that would help the reader understand what is going on.

(a) The abstract jumps straight in with jargon.

(b) The final two sentences of the abstract do not add anything helpful.

(c) A part of the introduction repeats the abstract word-for-word

(d) Detailed proofs of the fact that $a/(b+f(x))$ is monotonic in x if f is trivially monotonic in x are not

- needed. All of the derivations in the SI could be simplified by better use of compact notation, or removed entirely (note, some need to be changed anyway due to the issues with the model).
- (e) The authors need to come up with some names for their strategies that are less unwieldy, and easier to remember.
- (f) The sentence in lines 193-195, which is crucial, is very hard to understand. It might help if the appropriate domain were labelled in the associated diagram.
- (g) "transition free energy" sounds like a free-energy barrier.
- (h) The BM state is described as a kinetic trap, but at least as illustrated in Fig. 2 it's actually the thermodynamically favoured state.
- (i) It's not clear to me why the system of Figure 2 is needed in this paper. Why not go straight to the system of figure 3, if it works better? It's fairly obvious that it's going to be extremely slow as a detection mechanism. Indeed, in the limit that the authors want to use (no toehold for the exchange in reactions 5/6), exchange to the PM₁ state would also be excruciatingly slow.
- (j) The sudden switch to the new section at 386 is disorientating. What is DF? What is "low variation frequencies"? Why do F_S and F_B suddenly have new subscripts relative to the figure? Is this just about normalising and subtracting baselines, and if so, why isn't it just in the methods sections? Does the discussion apply to the previous results as well as the subsequent ones?
- (k) Figure 5b doesn't seem to show results that can be compared to those from sequencing (line 470).
- (l) In the SI, the difference between sections 2.4 and 2.5 is really hard to see from their names.
- (m) It is confusing that the authors use E and G for free energies. Why not use just G with appropriate superscripts and subscripts?

Minor points:

3. What does "thermodynamic inverse" mean?
4. What is the evidence that G-containing mismatches cause only slight changes to the thermodynamics? According to the Santalucia model (nupack) at least, most result in a few kT relative to the perfect duplex.
5. Z is introduced without definition (but in any case, needs to be eliminated if the modelling is done properly).
6. It is not very nice to have whole English words formatted in maths font in equations.
7. Lines 148-149 are not easy to understand.
8. Lines 180-181 are not easy to understand. What does it mean that the blocker does not "participate"?
9. It is not clear how you are changing the length of the blocker so that $\Delta\Delta G_{\text{diss}}$ is constant (lines 195-196). I think I worked out how (shortening it from the left hand edge), but this needs to be clearer.
10. One of the strands changes colour on the far RHS of fig. 2.
11. The labelling of the curves in fig. 3 is sub-optimal, since many cross or are in close proximity near $t \rightarrow \infty$.
12. Line 436 - why isn't the upgraded system of fig. 3 outperforming the system of fig. 2 in this setting? Or am I missing something?
13. Fig 4a. Why are these values taken at a time point before the fluorescence has reach saturation? Surely you need to wait for some kind of steady state?

14. The authors should probably cite Dave Zhang's X probes when they discuss the similar Y probes on line 460.

15. Methods - what were the slit widths?

Reviewer #2:

Remarks to the Author:

Manuscript introduces significantly improved method for specific and sensitive nucleic acid hybridization. Comprehensive theoretical model is presented and verified by experimental results on several examples. Approximations and limitations of the model are discussed. Mutation detection as a function of oligonucleotide concentrations and lengths are studied and agree well with the theoretical model. These relationships have not been previously published and are not obvious. Clinical biological application of this assay design is experimentally demonstrated. I have very little negative to say about the paper. It is well written and significant contribution to the field.

Minor issues:

- 1) While IF values are defined in the text DF values are introduced on line 387 without definition of abbreviation.
- 2) Origin of equation (7) on line 126 could be explained.
- 3) It would be useful to add dyes and quenchers within oligonucleotide sequences of Table S2. Locations of dyes and quenchers would make it easier to understand examples and analysis.
- 4) Some spelling errors. For example, "ideal gas constant" on line 115.
- 5) Epm symbol is not formatted well in the center of Figure 1, Figure S2-2, etc., in the pdf file.

Reponses to the reviewers

We would like to thank all the reviewers for the valuable comments and suggestions, and we have carefully revised our manuscript with all changes highlighted in red.

In general, we revised the manuscript on six major points:

- As was suggested by reviewer 1, we came up with the name “4-way **Strand-Exchange LEd Competitive DNA Testing system (4-Way SELECT system)**” for our system.
- After careful inspection of the statistical mechanics model adopted in the previous manuscript, we agree with reviewer 1 that the mass-action equilibria were more appropriate for our system. Therefore, we have adopted them to construct the theoretical model in the revised manuscript. Overall, the revised modelling results were similar to previous modelling results, but they were based on more reasonable approximations and easier to understand.
- The rationality of the approximations adopted in the theoretical model are directly demonstrated by comparing the analytic curves bearing approximations with the scatterplots of accurate solutions of mass-action equilibria equations with all parameters known and introduced.
- The mathematics of the whole theoretical model has been greatly simplified by a defined function of $F(x)$, which is actually the sensitivity of a toehold probe alone toward targeting strands (the variable x is the equilibrium constant). Aside from simplifying the mathematics, $F(x)$ also provides a clear physical picture of the underlying reaction mechanisms and thereby discloses the essence of our system.
- Based on all the modelling and experimental results, the essential contributions of our system to the field has been discussed in a more detailed and clearer way in the revised manuscript.
- All points on the small errors and the writing of the paper have been revised. Necessary supplementary experiments were conducted.

The point-by-point responses to each comment are as follows (The key points are emphasized in underline or bold):

Reviewer #1:

1. Theoretical modelling.

The authors describe their system through the "state" of a single target strand, with its different bonding configurations assigned free energies. Formally, this works ok if all "states" correspond to distinct bonding configurations of a single copy of that strand with other strands that are in enormous excess. However, that is not the case in the authors' work:

(a) Frequently, the blocker strands are not in excess.

(b) More worryingly, the distinct "states" of the model for the later designs (Fig. 2, Fig. 3) are not distinguished by the confirmation of the target strand alone: PM2 and PM1 are differentiated by the state of another strand. The authors should instead go back to the fundamental mass-action equilibria, eg:

$$[AB]/[A][B] = \exp(-dG^0_{AB}/RT),$$

and derive the steady states properly.

Response: We appreciate these two questions that point to the essence of the model and greatly push us to understand our system more deeply. We would like to response to question (b) first because clearance of it will help us answer question (a).

For question (b):

In the previous manuscript, we adopted the statistical mechanics model to calculate the distribution of different states. However, we did make a mistake that we used the Boltzmann distribution equation, as it was only applicable for systems consisting of independent particles that did not interact with each other. In our system, the particles, i.e. the strands, would interact with each other through base pairing. Therefore, it was wrong to adopt Boltzmann distribution equation. Moreover, as was pointed out by the reviewer, if we describe the system through the states of a single target strand, there existed only three states, not the four states depicted in Figure 2 and 3. In fact, **the 4-Way SELECT system could be regarded as a canonical ensemble** because the reaction temperature, the number of strands and the reaction volume were constant. The distribution of a canonical ensemble was exactly the same to Boltzmann distribution in terms of the formula (but they are quite different in the physical modelling):

$$Z_n = \frac{e^{-\frac{E_n}{RT}}}{Z} \quad (R1)$$

Where, E_n was the energy level of a certain state (the concept of "energy level E" in the canonical ensemble model or the Boltzmann distribution model was different from the concept of "free energy G"), Z_n was the possibility of the system staying at an energy level of E_n . Z was the partition function of a canonical ensemble,

$$Z = \sum_n e^{-\frac{E_n}{RT}} \quad (R2)$$

Overall, in the 4-Way SELECT system, there were four states as depicted in Figure 2 and 3 (Actually, there existed a fifth state that all the four strands were single stranded, but the energy level of this state was extremely high and its possibility was extremely low. So we could neglect it from our calculation with very little influence), and the distribution of these four states calculated in the previous manuscript was correct, but we misinterpreted the physical model as Boltzmann distribution.

As for the fundamental mass-action equilibria model suggested by the reviewer, our model was actually equivalent to it. Calculation of the distribution of a canonical ensemble model or Boltzmann distribution required the values of energy levels of all possible states. In

our manuscript, we obtained the energy levels through equation 7 in the previous manuscript (herein, we numbered it as equation R3):

$$E_{BW} = \Delta G_{BW} - RT \ln([\text{Blocker}]_0) \quad (\text{R3})$$

Equation R3 was derived from the following process: **Take the system in Figure 1 for example**, after the system reaching to the equilibrium state,

$$\frac{[\text{BWT}]}{[\text{Blocker}][\text{WT}]} = K_{BW} = e^{-\frac{\Delta G_{BW}}{RT}} \quad (\text{R4})$$

$$\frac{[\text{PWT}]}{[\text{Probe}][\text{WT}]} = K_{PW} = e^{-\frac{\Delta G_{PW}}{RT}} \quad (\text{R5})$$

In the canonical ensemble model, the involved states were states of the whole system, not a single strand. But for the system in Figure 1, the occupancies of ground state, BM state and PM state could be represented by the percentage of [WT], [BWT] and [PWT] over [WT]₀. Therefore, using the equilibrium equations R4 and R5,

$$\begin{aligned} Z_{BW} &= \frac{e^{-\frac{E_{BW}}{RT}}}{1 + e^{-\frac{E_{BW}}{RT}} + e^{-\frac{\Delta G_{PW}}{RT}}} = \frac{[\text{BWT}]}{[\text{WT}] + [\text{BWT}] + [\text{PWT}]} \\ &= \frac{e^{-\frac{\Delta G_{BW}}{RT}} [\text{Blocker}][\text{WT}]}{[\text{WT}] + e^{-\frac{\Delta G_{BW}}{RT}} [\text{Blocker}][\text{WT}] + e^{-\frac{\Delta G_{PW}}{RT}} [\text{Probe}][\text{WT}]} \\ &= \frac{e^{-\frac{\Delta G_{BW}}{RT}} [\text{Blocker}]}{1 + e^{-\frac{\Delta G_{BW}}{RT}} [\text{Blocker}] + e^{-\frac{\Delta G_{PW}}{RT}} [\text{Probe}]} \end{aligned} \quad (\text{R6})$$

Similarly,

$$Z_{PW} = \frac{e^{-\frac{E_{PW}}{RT}}}{1 + e^{-\frac{E_{BW}}{RT}} + e^{-\frac{\Delta G_{PW}}{RT}}} = \frac{e^{-\frac{\Delta G_{PW}}{RT}} [\text{Probe}]}{1 + e^{-\frac{\Delta G_{BW}}{RT}} [\text{Blocker}] + e^{-\frac{\Delta G_{PW}}{RT}} [\text{Probe}]} \quad (\text{R7})$$

Combine the above two equations,

$$e^{-\frac{E_{BW}}{RT}} = e^{-\frac{\Delta G_{BW}}{RT}} [\text{Blocker}] \quad (\text{R8})$$

$$e^{-\frac{E_{PW}}{RT}} = e^{-\frac{\Delta G_{PW}}{RT}} [\text{Probe}] \quad (\text{R9})$$

Then,

$$E_{BW} = -RT \ln \left(e^{-\frac{\Delta G_{BW}}{RT}} [\text{Blocker}] \right) = \Delta G_{BW} - RT \ln([\text{Blocker}]) \quad (\text{R10})$$

$$E_{PW} = -RT \ln \left(e^{-\frac{\Delta G_{PW}}{RT}} [\text{Probe}] \right) = \Delta G_{PW} - RT \ln([\text{Probe}]) \quad (\text{R11})$$

Toward equation R10 and R11, we adopted an approximation that the equilibrium concentration of blocker and probe were close to their initial concentrations, i.e. $[\text{Blocker}] \approx [\text{Blocker}]_0$, $[\text{Probe}] \approx [\text{Probe}]_0$ (I will discuss this approximation later). Then, we had the expressions of the energy levels of E_{BW} and E_{PW} ,

$$E_{BW} = \Delta G_{BW} - RT \ln([\text{Blocker}]_0) \quad (\text{R3})$$

$$E_{PW} = \Delta G_{PW} - RT \ln([\text{Probe}]_0) \quad (\text{R12})$$

Since we were focusing on the blocker's effect, we fixed the probe sequence and its initial concentration, so E_{PW} was regarded as a constant. Then, we could obtain the variations of the system's sensitivity and specificity over $[\text{Blocker}]_0$ and ΔG_{BW} . In short, for the system of Figure 1, we assigned energy levels for each state by equaling the canonical distribution to the thermodynamic equilibrium distribution. Therefore, these two distributions were actually equivalent.

For the 4-Way SELECT system, the occupancies of ground state, BM state, PM_1 state and PM_2 could be represented by the percentage of [MT], [BMT], [Ps] (the shorter strand of the probe, depicted in blue) and [BPs] over $[\text{MT}]_0$. Then, by following a similar derivation process, in which we equaled the distribution of canonical ensemble to the thermodynamic equilibrium distribution, we obtained,

$$E_{PM_1} = \Delta G_{PM_1} - RT \ln \left(\frac{([\text{Probe}]_0 - [\text{PMT}])}{[\text{PMT}]} \right) \quad (\text{R13})$$

$$E_{PM_2} = \Delta G_{PM_1} + \Delta G_{BM} - RT \ln \left(\frac{([\text{Probe}]_0 - [\text{PMT}])}{[\text{PMT}]} \right) - RT \ln[\text{Blocker}] \quad (\text{R14})$$

$$E_{BM} = \Delta G_{BM} - RT \ln([\text{Blocker}]) \quad (\text{R15})$$

$$\text{Sensitivity} = \frac{[\text{PMT}]}{c_0} = \frac{e^{-\frac{E_{PM_1}}{RT}} + e^{-\frac{E_{PM_2}}{RT}}}{1 + e^{-\frac{E_{BM}}{RT}} + e^{-\frac{E_{PM_1}}{RT}} + e^{-\frac{E_{PM_2}}{RT}}} = \frac{e^{-\frac{\Delta G_{PM_1}}{RT}} ([\text{Probe}]_0 - [\text{PMT}])}{e^{-\frac{\Delta G_{PM_1}}{RT}} ([\text{Probe}]_0 - [\text{PMT}]) + [\text{PMT}]} \quad (\text{R16})$$

Then,

$$\left(1 - e^{-\frac{\Delta G_{PM_1}}{RT}} [\text{Probe}]_0 \right) [\text{PMT}]^2 + \left(e^{-\frac{\Delta G_{PM_1}}{RT}} [\text{Probe}]_0 + c_0 e^{-\frac{\Delta G_{PM_1}}{RT}} \right) [\text{PMT}] - c_0 e^{-\frac{\Delta G_{PM_1}}{RT}} [\text{Probe}]_0 = 0 \quad (\text{R17})$$

After solving equation R17, we could obtain the expressions of [PMT] and Sensitivity over [Probe]₀ and ΔG_{PM}. Since we were focusing on the blocker's effect, we fixed [Probe]₀ and ΔG_{PM}. Then,

$$[PMT] \equiv \text{constant}, \text{ Sensitivity} \equiv \text{constant}, E_{PM_1} \equiv \text{constant}, [Probe] \equiv \text{constant}$$

Therefore, using the canonical ensemble distribution, we could draw a correct conclusion that the sensitivity of the 4-Way SELECT system was constant. However, we made mistakes in calculation of the system's specificity: we took it for granted that the differences in energy levels of different states were identical to the differences in free energies, and thereby we obtained equation 21 ($E_{PW1} - E_{PM1} = G_{PW1} - G_{PM1} = \Delta\Delta G_P$) in the previous manuscript. This was okay for the system in Figure 1, as E_{PM} , E_{BM} , E_{PW} and E_{BW} were only influenced by the corresponding free energies if the approximations of $[Blocker] \approx [Blocker]_0$ and $[Probe] \approx [Probe]_0$ were adopted. Whereas, in the 4-Way SELECT system, E_{PM1} , E_{PM2} , E_{PW1} and E_{PW2} were not only decided by the corresponding free energies even if the approximations of $[Blocker] \approx [Blocker]_0$ and $[Probe] \approx [Probe]_0$ were adopted. Of course, we could strictly follow the equations R13 to R15 and still use the canonical ensemble model to calculate for the 4-Way SELECT system, but it would actually increase our calculation burden because transforming the equilibrium equations into energy levels made the formula even more complex.

To sum up for the above discussion:

(1) The canonical ensemble model we adopted in the previous manuscript was equivalent to the mass-action equilibria model because we assigned energy levels for each state by equaling the canonical distribution to the thermodynamic equilibrium distribution.

(2) The canonical ensemble model itself did not require the concentration of blocker or probe to be in large excess. We adopted the approximations of $[Blocker] \approx [Blocker]_0$ and $[Probe] \approx [Probe]_0$ for the canonical ensemble model in order to make the mathematics more concise and easier to understand.

(3) The canonical ensemble model increased the calculation burden for the 4-Way SELECT system, and the physical picture of the whole system provided by it was complicated and confusing.

Overall, we agree with the reviewer's suggestion that the fundamental mass-action equilibria are more appropriate for the 4-Way SELECT system. So, we have adopted them to construct the theoretical models in the revised manuscript and SI.

For question (a):

As was mentioned above, in the previous manuscript, we adopted the approximations of $[Blocker] \approx [Blocker]_0$ and $[Probe] \approx [Probe]_0$ to simplify the calculation process, and we actually had discussed the rationality of such approximations in the discussion section of the previous manuscript: In the experiment concerning the influence of the blocker length on the system's specificity, $[Blocker]_0$ was set at 8-fold of the targeting strand in the experiments, which could be regarded as large excessive and would only cause minor deviations. As for the influence of the blocker concentration on the system's specificity, the concentrations used were in very large

excess at the end, which assured an accurate prediction of the upper limit of specificity. Though in the beginning, the blocker concentrations were not very large and might cause deviations in calculation, the specificity values under such conditions were not important because we would only apply the 4-Way SELECT system when its specificity was near the upper limit, around which the approximation of $[Blocker] = [Blocker]_0$ was quite close to the fact. **Furthermore, in the revised manuscript, we modified this approximation as $[Blocker] = [Blocker]_0 - c_0$** . This was more reasonable because almost all of the MT or WT had hybridized with blocker in the end. As for the approximation of $[Probe] = [Probe]_0$, which was adopted for system of Figure 1 and S1, we set the $[Probe]_0$ at 4-fold of the targeting strands in the computation, and when $[Blocker]_0$ was in large excess, most of the targeting strands would bind to Blocker, leaving the probes free. Therefore, similar to the approximation of $[Blocker] = [Blocker]_0$, the deviations caused by the approximation of $[Probe] = [Probe]_0$ at the early stage was considerable but not our interest, and at the end, the deviation would be very small and neglectable. We also would like to point out that for our 4-Way SELECT system, only the approximation of $[Blocker] = [Blocker]_0 - c_0$ was adopted.

In the revised manuscript, we used the **mass-action equilibria** to construct the models. **For system in Figure 1,**

$$[PMT] = \frac{[Probe]K_{PM}c_0}{[Blocker]K_{BM} + [Probe]K_{PM} + 1} \quad (R18)$$

$$[PWT] = \frac{[Probe]K_{PW}c_0}{[Blocker]K_{BW} + [Probe]K_{PW} + 1} \quad (R19)$$

$[Probe]$ and $[Blocker]$ were the equilibrium concentrations of probe and blocker, and they were all variables over $[Blocker]_0$. We proved that the sensitivity was monotonically decreasing with $[Blocker]$ and K_{BM} (see line 96-100 in the revised SI). Since,

$$[Blocker] \in [0, +\infty)$$

$$K_{BM} \in [0, +\infty)$$

Therefore,

$$\text{Sensitivity} = \frac{[PMT]}{c_0} \in \left[\frac{[Probe]K_{PM}}{[Probe]K_{PM} + 1}, 0 \right) \quad (R20)$$

However, for specificity, the calculation was very complicated. Since MT and WT reacted with probes and blockers in different tubes, the equilibrium concentrations of probe and blocker ($[Probe]$ and $[Blocker]$) were different. Therefore, we cannot calculate specificity through equations R18 and R19. Analytic expressions of $[PMT]$ and $[PWT]$ over $[Blocker]_0$ and $[Probe]_0$ must be obtained. We used Maple 2019 to calculate, but the output results were extremely long and complicated with a lot of nestification. It was nearly impossible to figure out the analytic expressions, let alone analyzing their monotonicity and limits. As a result, we still had to introduce some approximations for $[Blocker]_0$ and $[Probe]_0$,

$$[Blocker] = [Blocker]_0 - c_0 \quad (R21)$$

$$[\text{Probe}] = [\text{Probe}]_0 \quad (\text{R22})$$

We then could obtain much simpler analytic expressions of sensitivity and specificity,

$$\text{Sensitivity} = \frac{[\text{Probe}]_0 K_{PM} c_0}{[\text{Blocker}]_0 K_{BM} + [\text{Probe}]_0 K_{PM} + 1} \in \left[\frac{[\text{Probe}]_0 K_{PM}}{[\text{Probe}]_0 K_{PM} + 1}, 0 \right) \searrow (\text{R23})$$

$$\text{Specificity} = \frac{K_{PM}}{K_{BM}} \times \frac{([\text{Blocker}]_0 - c_0) K_{BW} + [\text{Probe}]_0 K_{PW} + 1}{([\text{Blocker}]_0 - c_0) K_{BM} + [\text{Probe}]_0 K_{PM} + 1} \in \left[\frac{K_{PM}}{K_{BM}} \times \frac{[\text{Probe}]_0 K_{PW} + 1}{[\text{Probe}]_0 K_{PM} + 1}, \frac{K_{PM}}{K_{BM}} \times \frac{K_{PW}}{K_{PM}} \right) \nearrow (\text{R24})$$

Specifically, we took a designed probe (Probe-1) as a model, and obtained its thermodynamic parameters through NUPACK. We then could draw the three-dimensional heatmaps of sensitivity and selectivity over $[\text{Blocker}]_0$ and $-\Delta G_{BW}$ and the two-dimensional interception curves of sensitivity and specificity over $[\text{Blocker}]_0$ by setting $-\Delta G_{BW}$ at a fixed value. Furthermore, we introduced the parameters into the fundamental mass-action equilibria and obtained accurate numerical solutions without any approximations. Using the accurate solutions, we drew the scatterplots. Shown in Figure R1 (Figure 1d in the revised manuscript), we could see that as a whole, the approximated interception curve was close to the accurate scatterplots especially when $[\text{Blocker}]_0$ was high, demonstrating the rationality of the approximations.

Figure R1. The sensitivity and specificity of the Standard probe/blocker composition system over $[\text{Blocker}]_0$. Probe-1 was used as the model.

For the 4-Way SELECT system, the analytic expressions of [PMT] and sensitivity could be derived without any approximations at all (see details in line 339-383 in the revised SI).

$$[\text{PMT}] = \frac{K_{PM1}[\text{Probe}]_0 + K_{PM1}c_0 - \sqrt{K_{PM}^2[\text{Probe}]_0^2 - 2K_{PM1}^2[\text{Probe}]_0c_0 + K_{PM1}^2c_0^2 + 4K_{PM1}[\text{Probe}]_0c_0}}{2(K_{PM1} - 1)} \equiv \text{constant} \quad (\text{R25})$$

Define the sensitivity of a toehold probe alone toward targeting strand as F(x), in which the variable x was the equilibrium constant. Then,

$$F(x) = \frac{([\text{Probe}]_0 + c_0) \cdot x + [\text{P}_s]_0 - \sqrt{([\text{Probe}]_0 + c_0)^2 \cdot x^2 + (4 \cdot [\text{Probe}]_0c_0 + 2 \cdot [\text{Probe}]_0[\text{P}_s]_0 + 2 \cdot [\text{P}_s]_0c_0) \cdot x + [\text{P}_s]_0^2}}{2c_0(x-1)} \quad (\text{R26})$$

Also, if we define,

$$\mu = K_{BW}[\text{Blocker}] = e^{-\frac{\Delta G_{BW}}{RT}} [\text{Blocker}] \quad (\text{R27})$$

$$\mu_{BM} = K_{BM}[\text{Blocker}] = e^{-\frac{\Delta \Delta G_B}{RT}} \mu \quad (\text{R28})$$

Then, the equation R25 could be transformed into the following form,

$$[\text{PMT}] = c_0 F\left(\frac{\mu_{BM} K_{PBM} + K_{PM1}}{\mu_{BM} + 1}\right) = c_0 F(K_{PM1}) = c_0 F\left(e^{-\frac{\Delta G_{PM1}}{RT}}\right) \equiv \text{constant} \quad (\text{R29})$$

Similarly,

$$[\text{PWT}] = \frac{\mu(e^{-\frac{\Delta G_{PBW}}{RT}}([\text{Probe}]_0 + c_0) + [\text{P}_s]_0) + e^{-\frac{\Delta G_{PW1}}{RT}}([\text{Probe}]_0 + c_0) + [\text{P}_s]_0 - \sqrt{(\mu e^{-\frac{\Delta G_{PBW}}{RT}} + e^{-\frac{\Delta G_{PW1}}{RT}})([\text{Probe}]_0 + c_0) - [\text{P}_s]_0(\mu + 1))^2 + 4[\text{Probe}]_0(\mu - 1)(c_0 + [\text{P}_s]_0)(\mu e^{-\frac{\Delta G_{PBW}}{RT}} + e^{-\frac{\Delta G_{PW1}}{RT}})}}{2\mu(e^{-\frac{\Delta G_{PBW}}{RT}} - 1) + 2(e^{-\frac{\Delta G_{PW1}}{RT}} - 1)}} = c_0 F\left(\frac{\mu K_{PBW} + K_{PW1}}{\mu + 1}\right) = c_0 F\left(\frac{e^{-\frac{\Delta G_{BW} + \Delta G_{PBW}}{RT}} [\text{Blocker}] + e^{-\frac{\Delta G_{PW}}{RT}}}{e^{-\frac{\Delta G_{BW}}{RT}} [\text{Blocker}] + 1}\right) \quad (\text{R30})$$

[PMT] and the sensitivity were constant, and [PWT] was proved to be monotonically decreasing over μ (see details in line 382-392 in the revised SI). Therefore, the value ranges of sensitivity and specificity could be determined,

$$\text{Sensitivity} = F\left(e^{-\frac{\Delta G_{PM}}{RT}}\right) \equiv \text{constant} \quad (\text{R31})$$

$$\text{Specificity} \in \left[\frac{F\left(e^{-\frac{\Delta G_{PM1}}{RT}}\right)}{F\left(e^{-\frac{\Delta G_{PW1}}{RT}}\right)}, \frac{F\left(e^{-\frac{\Delta G_{PM1}}{RT}}\right)}{F\left(e^{-\frac{\Delta G_{PBW}}{RT}}\right)} \right] \nearrow \quad (\text{R32})$$

However, we could not judge the monotonicity of [PWT] over $[\text{Probe}]_0$ and K_{BW} due to the extremely complexity of the expressions. Therefore, we adopted the approximations of $[\text{Blocker}] = [\text{Blocker}]_0 - c_0$. Then,

$$\mu = e^{-\frac{\Delta G_{BW}}{RT}} ([\text{Blocker}]_0 - c_0) \quad (\text{R33})$$

$$\text{Specificity} = \frac{F\left(e^{-\frac{\Delta G_{PM1}}{RT}}\right)}{F\left(\frac{e^{-\frac{\Delta G_{BW} + \Delta G_{PBW}}{RT}} ([\text{Blocker}]_0 - c_0) + e^{-\frac{\Delta G_{PW1}}{RT}}}{e^{-\frac{\Delta G_{BW}}{RT}} ([\text{Blocker}]_0 - c_0) + 1}\right)} \quad (\text{R34})$$

$$\frac{\partial \text{Specificity}}{\partial [\text{Blocker}]_0} = \frac{d\text{Specificity}}{dF\left(\frac{K_{PBW}\mu + K_{PW1}}{\mu + 1}\right)} \times \frac{dF\left(\frac{K_{PBW}\mu + K_{PW1}}{\mu + 1}\right)}{d\mu} \times \frac{\partial \mu}{\partial [\text{Blocker}]_0} > 0 \quad (\text{R35})$$

$$\frac{\partial \text{Specificity}}{\partial (-\Delta G_{BW})} = \frac{d\text{Specificity}}{dF\left(\frac{K_{PBW}\mu + K_{PW1}}{\mu + 1}\right)} \times \frac{dF\left(\frac{K_{PBW}\mu + K_{PW1}}{\mu + 1}\right)}{d\mu} \times \frac{\partial \mu}{\partial (-\Delta G_{BW})} > 0 \quad (\text{R36})$$

Thus,

$$\text{Specificity}([\text{Blocker}]_0, -\Delta G_{BW}) \in \left[\frac{F\left(e^{-\frac{\Delta G_{PM1}}{RT}}\right)}{F\left(e^{-\frac{\Delta G_{PW1}}{RT}}\right)}, \frac{F\left(e^{-\frac{\Delta G_{PM1}}{RT}}\right)}{F\left(e^{-\frac{\Delta G_{PBW}}{RT}}\right)} \right) \nearrow \quad (\text{R37})$$

Similarly, taking a designed probe (Probe-4) as a model, we obtained the accurate curve of sensitivity by equation R31, approximated curve of specificity by equation R33 and the accurate scatterplot of specificity by direct solving the mass-action equilibria with all parameters known and introduced. Shown in Figure R2, the approximated curve was extremely close to the accurate scatterplot, demonstrating the rationality of the approximation.

Figure R2. The sensitivity and specificity of the dissociative 4-Way SELECT system over $[\text{Blocker}]_0$. Probe-4 was used as the model.

To sum up, in the revised manuscript, we adopted approximations of $[\text{Blocker}] \approx [\text{Blocker}]_0 - c_0$ and $[\text{Probe}] \approx [\text{Probe}]_0$ for conventional competitive systems shown in Figure 1 and S1. Whereas, only $[\text{Blocker}] \approx [\text{Blocker}]_0 - c_0$ was adopted for the analytic expression of the specificity of 4-Way SELECT system (the analytic expression for its sensitivity was accurate, without any approximations). Furthermore, the rationality of the approximation for our system was directly demonstrated by accurate numerical solutions of the mass-action equilibria. Especially when $[\text{Blocker}]_0$ and K_{BW} were large, which was just the reaction condition that we would use in real

applications, the deviation between approximated curve and accurate scatterplot was extremely small.

I have been through the mathematics myself and I believe that the conclusions they draw about the expected equilibria of the systems illustrated in figures 2 and 3 are wrong. In particular, the tendency of the sensitivity to 1/2 when the conditions of eq. 25,26 are satisfied is wrong. In the limit of a large excess of reporter and blocker, the sensitivity should tend towards 1, not 1/2. Note that, in fact, this is what happens in the graphs in figure 3 - a discrepancy on which the authors do not comment. I suspect the results derived for the specificity are also incorrect.

Indeed, I believe to some extent that the authors "miss the point" about how their system works. The target strands will tend to bind to the reporters even with an excess of blockers, and ~0 dG difference, because there is an entropic drive to transfer the blue strands from the reporter complex to a complex with the blocker. This is what allows the specificity to approach 1, but it will also have consequences for the selectivity.

Response: We firstly would like to explain the discrepancy between equation 22/27 and the values in Figure 3a. In fact, we made a mistake that the Z-axis of Figure 3a should be "Normalized sensitivity", not the actual sensitivity. Also, in Figure 1a and Figure 2a, the sensitivity were normalized values too. Since we were focusing on the blocker's influence, we set the sensitivity of probes in the absence of blocker ($[\text{Blocker}]_0=0$) at 1 and normalized other calculated sensitivities over it. For specificity, the normalized values were calculated by setting the upper limit as 1. To make it clearer, we have used the actual sensitivity values to draw the heatmaps in the revised manuscript.

Actually, if we look at the equation R25, we could see that it was not against the reviewer's opinion that in the limit of a large excess of reporter (probe) and blocker, the actual sensitivity should tend towards 1, even though the $\Delta G_{PM} \approx 0$. For example, the equation R25 could be transformed into the following form,

$$\text{Sensitivity} = \frac{2K_{PM}[\text{Probe}]_0 c_0 - \sqrt{(K_{PM}[\text{Probe}]_0 - K_{PM}c_0)^2 + 4K_{PM}[\text{Probe}]_0 c_0}}{c_0(K_{PM}[\text{Probe}]_0 + K_{PM}c_0 + \sqrt{(K_{PM}[\text{Probe}]_0 - K_{PM}c_0)^2 + 4K_{PM}[\text{Probe}]_0 c_0})} \quad (\text{R38})$$

Then,

$$\lim_{[\text{Probe}]_0 \rightarrow \infty} \text{Sensitivity} = 1 \quad (\text{R39})$$

If we fixed the sequence and concentration of probes (i.e. $[\text{Probe}]_0$ and K_{PM} were constant), we could claim that the system's sensitivity was constant over $[\text{Blocker}]_0$ and K_{BW} . But if $[\text{Probe}]_0$ and K_{PM} varied, the sensitivity would change accordingly, as was pointed out by reviewer 1 that the sensitivity could approach 1 but it would also have consequences for the specificity.

To understand the essence of the 4-Way SELECT system, we could go back to the F(x) we defined, which was the sensitivity of a toehold probe with an equilibrium constant of x when hybridizing with the targeting strands. As was described in the revised SI, [PMT] and [PWT] could be both expressed in the form of $F((K_1X+K_2)/(X+1))$,

$$[\text{PMT}] = c_0 F\left(\frac{\mu_{BM} K_{PBM} + K_{PM1}}{\mu_{BM} + 1}\right) \quad (\text{R39})$$

$$[\text{PWT}] = c_0 F\left(\frac{\mu K_{PBW} + K_{PW1}}{\mu + 1}\right) \quad (\text{R40})$$

For MT, $K_{PBM}=K_{PM1}$, then equation R39 could be further simplified as equation R29. Looking over equation R39 and R40, we could disclose the essence of the 4-Way SELECT system: The influence of blocker on the hybridization thermodynamics seemed to be integrated into the hybridization thermodynamics of the hypothetical toehold probe, and **the 4-Way SELECT system as a whole could be regarded as a single toehold probe alone with an equilibrium constant of $(K_1X+K_2)/(X+1)$** , which was a combination of K_1 and K_2 . Increasing the concentration and length of the blocker changed the intrinsic equilibrium constant of the hypothetical toehold probe from the upper limit of K_{PM1} and K_{PW1} to the lower limit of K_{PBM} and K_{PBW} . **Overall, the whole 4-Way SELECT system was equivalent to a single toehold probe with its equilibrium constants toward MT and WT regulated by blocker.**

Based on the above discussion, we could discuss the **real contributions** of our system to the field:

For de novo design of a probe/blocker competitive composition system: the 4-way SELECT system had distinct advantages over the conventional systems:

i) If we compared equations R31-R32 with equations R23-R24, we could see that the conventional competitive composition suffered from the inverse correlation over both blocker and probe, whereas the 4-Way SELECT system only suffered that over probe. Therefore, to reach to the optimal detection zone, in which sensitivity and specificity were both relatively high, the optimization process for the conventional system has to balance the influences of two sides, and thereby the burden of sequence design, the optimization of experimental conditions and the susceptibility toward environmental changes were in the order of n^2 . Whereas, for the 4-Way SELECT system, it only suffered from the inverse correlation over probe, so we just needed to optimize the probe design. **The optimization burden was in the order of n , much smaller than that of conventional systems.**

ii) More importantly, **the optimization burden of probe in the 4-Way SELECT system could be experimentally free.** In real applications, we certainly would use long and excessive blocker strands to make the system function near its best specificity. Under such set-ups, the whole 4-Way SELECT system was equivalent to a single toehold probe alone with equilibrium constants of K_{PBM} (identical to K_{PM1}) and K_{PBW} , and the corresponding sensitivity and specificity of the single toehold probe and the 4-Way SELECT system were $F(K_{PM1})$ and $F(K_{PM1})/F(K_{PBW})$. Since the function of $F(x)$ was derived without taking any approximations at all, $F(K_{PM1})$ and $F(K_{PM1})/F(K_{PBW})$ were totally accurate and could be used for computerized de novo design of the 4-Way SELECT system. **Overall, we did not need to conduct any experiments for de novo design of the 4-Way SELECT system, and if we don't regard computer work as optimization, the 4-Way SELECT system was optimization free.**

On the contrary, **for conventional competitive compositions, thermodynamic modelling and computer work were far not enough for eliminating the experimental optimization process**: First, the accurate analytic expressions of sensitivity and specificity over $[\text{Blocker}]_0$ and $-\Delta G_{\text{BW}}$ were too complicated to be obtained and thereby approximations must be adopted. Besides, the probe in the conventional competitive compositions were single-stranded, which would be more susceptible toward environmental changes such as ionic strength than double-stranded probe in our system. Overall, there would exist inevitable deviations between the experimental results and theoretical calculation. Shown in Figure R3a (Figure 1d in the revised manuscript), although the deviation was acceptable as a whole, it was very large within the system's optimal detection zone. More terribly, the sensitivity and specificity changed harshly around the balanced point. Therefore, shown in the inset, such deviations rendered the optimal

Figure R3. The sensitivity and specificity of Standard probe/blocker composition system (a) and Strand displacement probe/blocker composition system (b) over $[\text{Blocker}]_0$. Probe-1 and Probe-2 were the used as models respectively. Insets: zoom in on the optimal zones.

detection zone predicted by approximated curves considerably deflected from that predicted by accurate scatterplots. We would like to point out that the accurate solutions of equation 3-4 were based on a certain system design with all the parameters known and introduced into calculation. So they were only output values of one specific system and could not provide guidance for sequence design and optimization. For conventional Strand displacement probe/blocker composition system, the deviation was so large that the optimal detection zones predicted by approximated curves and accurate scatterplots were completely segregated (Figure R3b herein or Figure S1d in the revised SI). Overall, for conventional composition systems, thermodynamic modelling that must adopt approximations was not predict the system's performance within the optimal detection zone. Intensive and cumbersome experimental optimization was inevitable.

Second, as was thoroughly discussed in 3.1 in the previous manuscript, the thermodynamic equilibrium state for the conventional system was very difficult to reach due to the ultra-slow kinetics of dsDNA dissociation. For example, the blocker and probe with same lengths of 18-nt and same concentrations would produce a sensitivity around 0.5 according to the thermodynamic model. However, the dissociation rate of blocker off the targeting strand was extremely small, and thereby we could only observe a very low sensitivity (nearly 0) during an acceptable experiment duration of several hours, which was a very large discrepancy with the predicted value of ~ 0.5 . By contrast, the kinetics was not a problem for the 4-Way SELECT system, as was thoroughly discussed in line 287-315 in the revised manuscript. To sum up, the large deviations within the optimal detection zone in combination with the ultra-slow kinetics made the quantitative prediction for conventional competitive composition systems extremely inaccurate, and thereby intensive experimental optimization was inevitable. Whereas, owing to the unique thermodynamics and fast kinetics, de novo design of the 4-Way SELECT system could be totally computerized and experimentally free.

iii) Since the analytic expressions of the sensitivity and the upper limit of specificity of the 4-Way SELECT system were completely accurate with no approximations at all, **empirical rules could further eliminate the computer work to make the 4-Way SELECT system rigorously optimization free.** Previously published papers¹ have reported that for systems consisting of only probes and MT/WT, the near-optimal performance could be achieved simply by setting $\Delta G_{PM} \approx 0$. In real applications, the 4-Way SELECT system was equivalent to a system consisting of only toehold probe and MT/WT. Therefore, if we don't rigorously pursue the most optimal performance, such empirical rule of $\Delta G_{PM} \approx 0$ could be applied to our system to make it completely optimization free, as was experimentally demonstrated in Figure 5. We also would like to point out that for dissociative 4-Way SELECT system described in Line 340-359 in the revised manuscript, $\Delta G_{PM} \equiv 0$, so it was set to be totally optimization free by default.

For enhancing the performance of detection systems that already comprised probes:

In many cases, researchers may have designed and synthesized strand displacement probes in their assays. But the experimental results, especially the specificity, were not satisfying. To improve the specificity, addition of a blocker strand into the system would be the first choice. In such situations, blockers, designed in the principle of our 4-Way SELECT system, showed huge advantages over other probe/blocker systems: conventional blocker would immediately lower down the system's sensitivity when increasing its specificity. By contrast, our

proposed blocker would only increase the specificity to its upper limit without affecting the sensitivity at all, which was exceedingly more convenient to employ. Especially for the enhancement of detection systems targeting multiple mutations of interests, our system would be the best choice.

To sum up, the de novo design of a 4-Way SELECT system was free of experimental optimization, and it could be completely optimization-free by following some empirical rules (e.g. let the length of the docking domain to be close to the length of the dissociation domain). The proposed 4-Way SELECT system also provided a very convenient tool for the enhancement of genetic testing systems that already comprised probes.

2. The writing of the paper.

(a) The abstract jumps straight in with jargon.

Response: We have re-written the abstract to make it plainer and easier to understand.

(b) The final two sentences of the abstract do not add anything helpful.

Response: We have removed these two sentences in the revised manuscript.

(c) A part of the introduction repeats the abstract word-for-word

Response: We have re-written the last paragraph of the introduction to make it more concise.

(d) Detailed proofs of the fact that $a/(b+f(x))$ is monotonic in x if f is trivially monotonic in x are not needed. All of the derivations in the SI could be simplified by better use of compact notation, or removed entirely (note, some need to be changed anyway due to the issues with the model).

Response: We have revised the thermodynamic model by using the fundamental mass-action equilibria, and by defining functions of $F(x)$ and $G(x)$, the mathematics were greatly simplified while the physical picture behind the mathematics became much clearer instead.

(e) The authors need to come up with some names for their strategies that are less unwieldy, and easier to remember.

Response: We have come up with the name “4-way Strand-Exchange LE_d Competitive DNA Testing system (4-Way SELECT system)” for our strategy.

(f) The sentence in lines 193-195, which is crucial, is very hard to understand. It might help if the appropriate domain were labelled in the associated diagram.

Response: In the previous manuscript, $\Delta\Delta G_{\text{dis}}$ was defined as the difference between the free energy change of domain a and that of domain b.

However, in the revised manuscript, we used mass-action equilibria to construct the thermodynamic model, and the concept of $\Delta\Delta G_{\text{dis}}$ was no longer needed.

(g) "transition free energy" sounds like a free-energy barrier.

Response: We have changed "transition free energy" to "free energy change" in the revised manuscript.

(h) The BM state is described as a kinetic trap, but at least as illustrated in Fig. 2 it's actually the thermodynamically favoured state.

Response: In real application, the workflow of a probe/blocker composition system was usually divided into two steps. First, the MT or WT would adequately react with blocker. Then, the probe was introduced for subsequent reaction. In figure 2, the BM state was indeed a thermodynamically favored state. Therefore, most of MT would transform into BMT after the first step. After addition of probes, reaction 1-6 were expected to happen and the established thermodynamic model described the global equilibrium state of reactions 1-6. Since the vast majority of the species in the system was BMT at the beginning of step 2, reaching to the global equilibrium state relied on fast kinetics of reaction 2. However, the kinetic constant of reaction 2 was extremely small ($\sim 10^{-35}$), especially when $[\text{Blocker}]_0$ and $-\Delta G_{\text{BW}}$ were large. Therefore, BMT dissociated very slowly and the mutant strand was "kinetically trapped" at the BM state. In the end, the whole reaction system could not reach to global thermodynamic equilibrium within a typical reaction time of several hours, and the vast majority of the species in the solution was still BMT, which seemed that the whole reaction system was "trapped" at the BM state. To sum up, the BM state was described as a kinetic trap because the mutant strand and the whole reaction system was trapped at the BM state and could not escape away due to the ultra-small kinetic constant of reaction 2.

(i) It's not clear to me why the system of Figure 2 is needed in this paper. Why not go straight to the system of figure 3, if it works better? It's fairly obvious that it's going to be extremely slow as a detection mechanism. Indeed, in the limit that the authors want to use (no toehold for the exchange in reactions 5/6), exchange to the PM₁ state would also be excruciatingly slow.

Response: The thermodynamic model of the system of Figure 2 indicated that although the sensitivity and specificity herein were still inversely correlated, the sensitivity, unlike the conventional probe/blocker composition system, no longer decreased to 0 when $[\text{Blocker}]_0$ and $-\Delta G_{\text{BW}}$ were very large. And in the limit that the length of the dissociation domain was 0 ($\Delta\Delta G_{\text{dis}} = 0$), the lower limit of sensitivity could be as high as 0.5, which assured a wide optimal zone for detection because we could simply increase the length and concentration of blocker

strand to enhance the system's specificity without the concern of too much loss of sensitivity. Briefly, the limit we used was not "no toehold for the exchange in reactions 5/6" but "the length of the dissociation domain was 0 ($\Delta\Delta G_{\text{dis}} = 0$)". The short length of the dissociation domain would not make the exchange to PM_1 state slower. Also, we would like to point out that for the mass-action equilibria base model in the revised manuscript, the concept of $\Delta\Delta G_{\text{dis}}$ was no longer used.

In the experiment, we observed that the system's sensitivity decreased to nearly 0 when $[\text{Blocker}]_0$ and $-\Delta G_{\text{BW}}^{\emptyset}$ were large, which was in contradiction to calculation. Through analysing the reaction pathways, we found that the discrepancy was derived from the fact that the thermodynamic model required the reactions 1-6 to reach to global equilibrium whereas the ultra-small kinetic constant of reaction 2 impeded the equilibrium process.

Based on the above kinetics analysis, we came up with the idea of creating a new pathway for MT/WT to escape from the BM state. After careful inspection of the reactions in the system of Figure 2, we thought that the best way was to create a reaction pathway between BM state and PM_2 state. In the current design, there was no toehold domain to initiate the transition from BM state to PM_2 state. Quite naturally, we had the idea of moving away the blocker sequence to spare out a toehold domain, and thereby MT/WT could escape from the BM state to PM_2 state through Holliday junction branch migration (4-way strand exchange). And **this was how the kinetics of the system of Figure 2 guided us to the invention of 4-Way SELECT system**. Moreover, the value range of the system's sensitivity was as follows (equation 17 in the revised manuscript, herein, we numbered it as equation R41),

$$\text{Sensitivity}([\text{Blocker}]_0, -\Delta G_{\text{BW}}) = F\left(\frac{K_{PM1}K_{BP_s}}{K_{BM}} \mu_{BM} + K_{PM1}\right) \in \left[F\left(e^{\frac{\Delta G_{PM1}}{RT}}\right), F\left(e^{\frac{\Delta G_{PM1} - \Delta G_{BM} - \Delta\Delta G_B + \Delta G_{BP_s}}{RT}}\right)\right] \quad (R41)$$

The value range provided important guidance that if we move the blocker sequence to an appropriate domain, ΔG_{BM} (i.e. $\Delta G_{\text{BW}} + \Delta\Delta G_B$) could be identical to ΔG_{BP_s} and the system's sensitivity could be constant. **This was how the thermodynamics of the system of Figure 2 guided us to the invention of 4-Way SELECT system.**

Taking the guidance together, we moved the blocker sequence to be restrained within the branch migration domain. Thus, the sequences of the hybridization domains of BPs and BMT were exactly the same, which guaranteed $\Delta G_{\text{BM}} = \Delta G_{\text{BP}_s}$. As for the kinetics, restriction of the blocker within the branch migration domain spared out a single stranded domain in BMT for initiating the four-way strand exchange process. It was worth noting that the 4-way SELECT system was still not able to reach to global thermodynamic equilibrium because the kinetic constants of reaction 2 and 6 were very small. So the vast majority of the species were still trapped at the BM state and PM_2 state after several hours. As was discussed in the previous manuscript and the revised manuscript (line 287-305), the BM state and PM_2 state could reach to localized equilibrium between each other through 4-way strand exchange. Fortunately, the free energies of BM state and PM_2 state were significantly lower than the other two states. So, at the final global equilibrium state, almost all of the species were distributed into the BM state and PM_2 state. Thus, the localized equilibrium between BM state and PM_2 state could be very close to the global equilibrium. Overall, the 4-Way SELECT system was still not able to reach

to the final global equilibrium, but owing to the 4-way strand exchange, it could quickly reach to the localized equilibrium that was very close to the final global equilibrium. Therefore, the established thermodynamic model, although based on global equilibrium, could predict the system's performance at the localized equilibrium state with little deviations.

To sum up, guided by the thermodynamics of the system of Figure 1, we invented the system of Figure 2. And guided by both the thermodynamics and the kinetics of the system of Figure 2, we finally invented the 4-way SELECT system that could break up the inverse correlation between sensitivity and specificity. We believe that the system of Figure 2 to some extent disclosed the vital kinetic mechanism of a competitive probe/blocker system, which might be scientifically sound to many researchers. Moreover, it clearly showed how the kinetics guided us to the invention of 4-way SELECT system and could help readers understand more clearly on the localized equilibrium and the global equilibrium in our system. **We thereby chose to retain Figure 2 in the revised manuscript, but we have explained the kinetic and thermodynamic guidance provided by the system of Figure 2 in a more detailed and clearer way.**

(j) The sudden switch to the new section at 386 is disorientating. What is DF? What is "low variation frequencies"? Why do F_S and F_B suddenly have new subscripts relative to the figure? Is this just about normalising and subtracting baselines, and if so, why isn't it just in the methods sections? Does the discussion apply to the previous results as well as the subsequent ones?

Response: DF was the abbreviation for discrimination factor, and it was defined as the ratio of the signal produced by a certain amount of MT to that by WT with identical amount. Therefore, it was an experimental value that directly represented the system's specificity.

Variation frequency was defined as the percentage of MT that carried genetic variations in a mixture of MT and WT. In real applications, especially cancer diagnosis, only a small fraction of cells in tumour tissues carried the targeting genetic variation while most of the cells were wild-type. Therefore, the genetic DNA samples extracted from tumour tissues were mostly wild-type DNAs with low variation frequency. The term "low variation frequency" was also used in previously published papers^{2,3}.

As for the discrepancy between the subscripts in equation 30 and Figure 4 in the previous manuscript, it was our mistake to mix them up, we have changed the subscripts in equation 30 to make it in accordance with Figure 4. As was discussed in the previous manuscript, the detection of low-frequency mutations, which was very important for molecular diagnosis of cancers, required the method to recognize small amount of MT in a large background of WT. In real detection, the signal of the unknown sample was compared with that of the negative control (pure WT), and the larger difference between their signals, the better detection performance we could achieve. Herein, the numerator in the definition of IF (i.e. F_a) was defined as the difference of fluorescence intensity between the negative control and the sample with a mutation frequency of VAF. F_a could actually be regarded as the "Signal" of the detection. The denominator in the definition of IF (i.e. F_b) was defined as the difference of fluorescence intensity between the negative control and the blank control. Thus, F_b could be regarded as the

“Background” of the detection. Taken together, IF represented the Signal-to-background ratio of a detection system toward samples with a mutation frequency of VAF. Surely, IF was a much more accurate and straightforward parameter to reflect the ability of our 4-Way SELECT system in detecting low frequency mutations.

We also would like to point out that the hybridization efficiencies of probes toward different concentrations of targeting strands were not rigorously proportional, especially when the concentration of targeting strands varied by 2 or more orders. Therefore, the IF value would change along with the mutation frequencies (lower mutation frequencies meant lower concentrations of MT when the total amount of DNA was fixed), which was actually beneficial for accurately assessing the system’s performance under different situations. Whereas, the parameter “specificity” was constant and thereby not suitable for such assessment, especially when the mutation frequencies were relatively low.

For the calculation of DF, the signal of MT and WT were both subtracted by the fluorescence intensity of the blank control, and this did apply to the previous and subsequent results on specificity. But the IF discussed herein could not simply apply to previous results or the subsequent ones. The IF was experimentally derived for reflecting the system’s ability in low-frequency mutation detection while the DF was experimentally derived for measuring the system’s specificity. Toward different applications, we should adopt the appropriate parameter accordingly.

Since DF directly represented the system’s specificity and the calculation of specificity in previous and subsequent results were exactly the same as the calculation of DF, we chose to eliminate the term “DF” and use “specificity” to make the manuscript more coherent. Also, we have discussed the above points in the revised manuscript to make it clearer and more straightforward.

(k) Figure 5b doesn't seem to show results that can be compared to those from sequencing (line 470).

Response: For the figure legends in Figure 5b and Figure S8, we changed them from “Mutant type” and “Wild type” to “Ovarian cancer patient” and “Healthy volunteer” so that the fluorescent signal could be corresponded to the sequencing results.

(l) In the SI, the difference between sections 2.4 and 2.5 is really hard to see from their names.

Response: As was mentioned in question (e), we named our system as 4-Way SELECT system, and accordingly, the name for section 2.4 was changed to “Dissociative 4-Way SELECT system” and the name for section 2.5 was changed to “Non-dissociative 4-Way SELECT system”.

(m) It is confusing that the authors use E and G for free energies. Why not use just G with appropriate superscripts and subscripts?

Response: Since we have adopted the mass-action equilibria to construct the theoretical models, we did not need the concept of “Energy lever (E)” anymore. So we used G with appropriate superscripts and subscripts in the revised version.

Minor points:

3. What does "thermodynamic inverse" mean?

Response: We intended to use "thermodynamic inverse" as a concise form of the concept of "inverse correlation between the sensitivity and specificity in thermodynamics". To make it more precise, we changed the term to "inverse correlation in thermodynamics"

4. What is the evidence that G-containing mismatches cause only slight changes to the thermodynamics? According to the Santalucia model (nupack) at least, most result in a few kT relative to the perfect duplex.

Response: G:X mismatches could indeed result in a few kT relative to the perfect duplex. The slight thermodynamic changes we claimed was relative to the changes caused by other types of mismatches such as X:C, and this was also demonstrated or claimed by many published papers⁴⁻⁷. To make it more clear, we have revised the texts as "especially for stable mismatches such as G:X (X=A,T,G) which cause considerably slighter changes to the thermodynamics as compared to other types of mismatches such as C:X, leading to low discrimination efficiency".

5. Z is introduced without definition (but in any case, needs to be eliminated if the modelling is done properly).

Response: Z_n was the possibility of the system staying at an energy level of E_n . Z was the partition function of a canonical ensemble. As was mentioned above, we have adopted the mass-action equilibria to construct the theoretical model in the revised version, so Z was eliminated.

6. It is not very nice to have whole English words formatted in maths font in equations.

Response: We have revised the formats according to the reviewer's suggestion.

7. Lines 148-149 are not easy to understand.

Response: The texts in lines 148-149 were revised as "Typically, researchers expected that the sensitivity and specificity of a detection system were both relatively high. Therefore, we defined an optimal function zone for a detection system, within which the system's sensitivity and specificity were both higher than 80% of the system's sensitivity and specificity at the balanced point. Marked in grey in Figure 1d, the optimal function zone of the conventional probe/blocker composition systems was very narrow." (line 142-146 in the revised manuscript)

8. Lines 180-181 are not easy to understand. What does it mean that the blocker does not "participate"?

Response: The texts in lines 180-181 were revised as "the key reason for the monotonic decreasing of sensitivity was that the blocker strand was completely a negative factor to the system's sensitivity as it could not provide any free energy drive for the formation of PMT at all. Herein, we tried to optimize the probe's structure so that the blocker strand, aside from the

hybridization with MT, could in a way provide some free energy drive for the formation of PMT. We then came up with a novel probe design approach: strand displacement probe/standard blocker composition system.” (line 184-189 in the revised manuscript)

9. It is not clear how you are changing the length of the blocker so that $\Delta\Delta G_{\text{dis}}$ is constant (lines 195-196). I think I worked out how (shortening it from the left hand edge), but this needs to be clearer.

Response: As was answered question 2(f), in the previous manuscript, $\Delta\Delta G_{\text{dis}}$ was the difference between the free energy change of domain a and that of domain b. Therefore, it was only determined by the structure of probe and targeting strand, and changing the length of the blocker had no influence on it.

However, in the revised version, $\Delta\Delta G_{\text{dis}}$ was no longer needed for the mass-action equilibria based thermodynamic model, so we did not add any explanations on this issue in the revised manuscript.

10. One of the strands changes colour on the far RHS of fig. 2.

Response: We have corrected this error.

11. The labelling of the curves in fig. 3 is sub-optimal, since many cross or are in close proximity near $t \rightarrow \infty$.

Response: We have labelled the curves in Figure 3 in a clearer way.

12. Line 436 - why isn't the upgraded system of fig. 3 outperforming the system of fig. 2 in this setting? Or am I missing something?

Response: After careful inspection of the previous manuscript, we guess that on this question the reviewer was referring to the texts in line 336 not those in line 436.

From the results displayed in Figure 2 and Figure 3, we could clearly see that the upgraded 4-Way SELECT system of Figure 3 greatly outperformed the system of Figure 2 theoretically and experimentally. Theoretically, the sensitivity of the system of Figure 2 decreased with the increase of $[\text{Blocker}]_0$. Whereas, the sensitivity of our 4-Way SELECT system remained unchanged. Experimentally, the sensitivity of the system of Figure 2 decreased to around 0 in the end and that of our 4-Way SELECT system was almost constant. Also, due to the decrease of $[\text{PMT}]$, the experimental specificity was significantly lower than that of our 4-Way SELECT system.

As for the discussion in line 336, we were actually referring to the limit of the instrument we used: although theoretical calculations predicted a very high specificity, the limitation of the fluorescence signal intensity that could be detected by the instrument, the inherent noise of the instrument and the inevitable errors alongside the whole detection process prevented us from detecting very slight fluorescent signals produced by little amount of PWT, thus setting an upper limit of specificity that could be accurately measured. In consideration of the signal window and

the standard deviations observed in the experiments, we estimated the upper limit of the specificity that could be accurately measured by the instrument was 250.

We have explained the experimental limitations in a more detailed way in the revised manuscript (line 329-335).

13. Fig 4a. Why are these values taken at a time point before the fluorescence has reach saturation? Surely you need to wait for some kind of steady state?

Response: We have re-conducted the experiments of Figure 4a and 4b and calculated the IF using the plateaued fluorescence intensities. The results were shown in the revised Figure 4a and 4b.

14. The authors should probably cite Dave Zhang's X probes when they discuss the similar Y probes on line 460.

Response: We have cited the paper in the revised manuscript.

15. Methods - what were the slit widths?

Response: The slit widths for excitation and emission were 485/20 nm and 528/20 nm respectively. We have added this in the revised manuscript.

Reviewer #2 (Remarks to the Author):

Manuscript introduces significantly improved method for specific and sensitive nucleic acid hybridization. Comprehensive theoretical model is presented and verified by experimental results on several examples. Approximations and limitations of the model are discussed. Mutation detection as a function of oligonucleotide concentrations and lengths are studied and agree well with the theoretical model. These relationships have not been previously published and are not obvious. Clinical biological application of this assay design is experimentally demonstrated. I have very little negative to say about the paper. It is well written and significant contribution to the field.

Minor issues:

1. While IF values are defined in the text DF values are introduced on line 387 without definition of abbreviation.

Response: As was thoroughly answered above (reviewer 1, question 2j), since DF directly represented the system's specificity and the calculation of specificity in previous and subsequent results were exactly the same as the calculation of DF, we chose to eliminate the term "DF" and use "specificity" to make the manuscript more coherent.

2. Origin of equation (7) on line 126 could be explained.

Response: We appreciate the reviewer for questioning the key equation in our model, which greatly pushes us to understand our system more deeply. The origin of equation (7) was thoroughly answered in question 1 raised by reviewer 1. And we found that the mass-action

equilibria suggested by reviewer 1 were more appropriate and straightforward for our system, so we adopted them to construct theoretical models in the revised manuscript. Overall, modelling results in the revised manuscript were similar to those in previous manuscript, but they were based on more reasonable approximations and easier to understand.

3) It would be useful to add dyes and quenchers within oligonucleotide sequences of Table S2. Locations of dyes and quenchers would make it easier to understand examples and analysis.

Response: We have added dyes and quenchers within the oligonucleotide sequences of Table S2 in the revised SI.

4) Some spelling errors. For example, “ideal gas constant” on line 115.

Response: We have checked through the whole manuscript and revised all spelling errors we have found.

5) Epm symbol is not formatted well in the center of Figure 1, Figure S2-2, etc., in the pdf file.

Response: In the revised manuscript, mass-action equilibria were used to construct the theoretical models. Therefore, the concept of E_{PM} was no longer used.

Reference

- 1 Wang, J. S. & Zhang, D. Y. Simulation-guided DNA probe design for consistently ultraspecific hybridization. *Nat Chem* **7**, 545-553, doi:10.1038/nchem.2266 (2015).
- 2 Haworth, S. *et al.* Low-frequency variation in TP53 has large effects on head circumference and intracranial volume. *Nat Commun* **10**, 357, doi:10.1038/s41467-018-07863-x (2019).
- 3 Hiatt, J. B., Pritchard, C. C., Salipante, S. J., O'Roak, B. J. & Shendure, J. Single molecule molecular inversion probes for targeted, high-accuracy detection of low-frequency variation. *Genome Res* **23**, 843-854, doi:10.1101/gr.147686.112 (2013).
- 4 Allawi, H. T. & SantaLucia, J. Thermodynamics and NMR of internal GT mismatches in DNA. *Biochemistry* **36**, 10581-10594, doi:10.1021/bi962590c (1997).
- 5 Lucarelli, F., Marrazza, G. & Mascini, M. Design of an optimal allele-specific oligonucleotide probe for the efficient discrimination of a thermodynamically stable (G-T) mismatch. *Analytica Chimica Acta* **603**, 82-86, doi:10.1016/j.aca.2007.09.047 (2007).
- 6 Piao, X., Sun, L., Zhang, T., Gan, Y. & Guan, Y. Effects of mismatches and insertions on discrimination accuracy of nucleic acid probes. *Acta Biochimica Polonica* **55**, 713-720 (2008).
- 7 Tikhomirova, A., Beletskaya, I. V. & Chalikian, T. V. Stability of DNA duplexes containing GG, CC, AA, and TT mismatches. *Biochemistry* **45**, 10563-10571, doi:10.1021/bi060304j (2006).

Reviewers' Comments:

Reviewer #1:

Remarks to the Author:

The authors have substantially revised their manuscript, and it is much improved. As before, based on the idea and results alone, I am minded to recommend acceptance in Nature Communications. However, there is still too much imprecision in the way the results are discussed and presented. I believe all of these issues can be fixed, but they really should be before publication can be approved.

1. The biggest problem is that, although the authors have taken my advice to use the mass action equilibria formulae, they continue to draw diagrams and write discussions that refer to "states" in a way that is not helpful.

It would make sense to talk about eg. the four "states" in figure 3, with precisely defined free energies, if the system consisted of only four strands in a box. But it does not, it consists of a bulk volume of strands, and bulk systems do not behave like small systems with the same density of strands. For example, see:

<https://iopscience.iop.org/article/10.1088/0953-8984/22/10/104102/meta>
<https://aip.scitation.org/doi/abs/10.1063/1.4757267>

It is ok to talk about the states of a single species if all partners with which it interacts are in large excess. Then, each strand can be viewed independently. If, however, concentrations of the other strands are finite, then the strands are coupled through total concentration conservation laws. Moreover, it makes even less sense if the "states" are distinguished by the configurations of different strands, as in Fig. 3a. There is no meaningful way to pair individual blue and red strands together to say whether a red strand is in the PM1 or PM2 "state" in a bulk system.

The above argument was the whole point of using the mass action formulae, along with the total concentration conservation laws, in the analysis. As the authors have seen, their results have changed. Therefore their discussion should also change. Instead of states, the authors should be referring simply to the complexes that exist in solution, the reactions that those complexes can undergo, and the associated standard free energy changes (defined at 1M of each reactant and product). The diagram with the four sets of strands is fine, provided that the authors stop labelling the sets of complexes as "states", and putting them on an absolute free energy scale.

The remaining points are more minor, but still worth addressing.

2. It is actually quite easy to prove monotonicity of sensitivity with respect to K_{BM} . Consider an increase in K_{BM} ; we want to prove that $[PMT]$ necessarily decreases. For contradiction, assume not ($[PMT] \geq [PMT_{old}]$).

- Thus by the mass action formula for PMT formation, $[MT] > [MT_{old}]$ since K_{PM} unchanged and $[Probe]$ decreases.

- Thus by the conservation equation $[MT] + [PMT] + [BMT] = c_0$, $[BMT] < [BMT_{old}]$.

- But $[BMT] < [BMT_{old}]$ ($\Rightarrow [Blocker] > [Blocker_{old}]$) and $[MT] > [MT_{old}]$ violates the mass action law for BMT formation, since K_{BM} has increased. Thus we have a contradiction and $[PMT]$ necessarily decreases.

It may be possible to make progress along the other lines of investigation using similar reasoning.

3. Variables appear without definition, or are defined without appearing, in numerous places. See, for example:

- R and T on line 101

- $[Ps]_0$. Is this the total concentration of the strand, or the initial concentration of free strand?

- The monotonically increasing/decreasing arrows $n_{w,s}$ to be defined. In addition, in eq. 12 and 13 it doesn't even say which quantity they increase or decrease with respect to.

- The K and X expression on line 283.

4. The "SELECT" name is a big improvement, but the systems described on lines 165 and 182 need to be more clearly distinguished.

5. $F(x)$ is an important function. It would make sense to write down the equations from which it follows. I think it would be better to explicitly describe it as a function of the concentrations as well as K (or x). This would really help, since the concentrations that go into F are different in different contexts.

6. What does quasi-monotonically mean?

7. I think "decreasing" on line 269 should be "increasing".

8. Personally, I think it would REALLY help if all strands were given a symbol, and complexes were represented by putting the two symbols together (eg. P, MT and PMT). The current system is confusing.

9. On the assumption of $[\text{Blocker}] = [\text{Blocker}] - c_0$.

(a) This is actually a different sort of assumption in different cases. In the original system, it is essentially an assumption that either blocker is more prevalent, or binds more strongly to the the target strand, than the probe. It is not, as the authors say in 435, an assumption that the blocker strand "changes very little". It changes as much as possible!

(b) In the SELECT context, the assumption is satisfied immediately if bonding of blocker to target or Ps is strong. There is no requirement that the blocker out compete the probe. It is likely to be an extremely good assumption in this setting.

(c) What do the authors mean by "in the beginning" on line 441? It sounds, from the previous sentence, like they are talking about the beginning of the experiment, but this wouldn't make sense.

Responses to the reviewers

We would like to thank the reviewer for the valuable comments and suggestions, and we have carefully revised our manuscript with all changes highlighted in red. As was suggested by the editors, we also edited the format of our manuscript to comply with the journal's requirements, e.g. we arranged the figures and figure legends at the bottom of the main article and deleted the subheadings of the discussion section.

The point-by-point responses to each comment are as follows:

The authors have substantially revised their manuscript, and it is much improved. As before, based on the idea and results alone, I am minded to recommend acceptance in Nature Communications. However, there is still too much imprecision in the way the results are discussed and presented. I believe all of these issues can be fixed, but they really should be before publication can be approved.

1. The biggest problem is that, although the authors have taken my advice to use the mass action equilibria formulae, they continue to draw diagrams and write discussions that refer to "states" in a way that is not helpful.

It would makes sense to talk about eg. the four "states" in figure 3, with precisely defined free energies, if the system consisted of only four strands in a box. But it does not, it consists of a bulk volume of strands, and bulk systems do not behave like small systems with the same density of strands. For example, see:

<https://iopscience.iop.org/article/10.1088/0953-8984/22/10/104102/meta>

<https://aip.scitation.org/doi/abs/10.1063/1.4757267>

It is ok to talk about the states of a single species if all partners with which it interacts are in large excess. Then, each strand can be viewed independently. If, however, concentrations of the other strands are finite, then the strands are coupled through total concentration conservation laws. Moreover, it makes even less sense if the "states" are distinguished by the configurations of different strands, as in Fig. 3a. There is no meaningful way to pair individual blue and red strands together to say whether a red strand is in the PM1 or PM2 "state" in a bulk system.

The above argument was the whole point of using the mass action formulae, along with the total concentration conservation laws, in the analysis. As the authors have seen, their results have changed. Therefore their discussion should also change. Instead of states, the authors should be referring simply to the complexes that exist in solution, the reactions that those complexes can undergo, and the associated standard free energy changes (defined at 1M of each reactant and product). The diagram with the four sets of strands is fine, provided that the authors stop labelling the sets of complexes as "states", and putting them on an absolute free energy scale.

Response: We really appreciate the reviewer for providing vital information on the thermodynamic difference between the bulk system and the small system, which helped us

describe our system more accurately. In the diagrams of the revised manuscript, we have labelled the sets of strands with the names of all the associated species, e.g. P + B + MT (As was suggested by the reviewer, we gave the probe strand a symbol P and blocker strand a symbol B). The absolute free energy scale was changed to the free energy change scale. Correspondingly, in the texts, we have used species that exist in solution, the reactions that those species can undergo, and the associated standard free energy changes to describe and discuss the systems.

2. It is actually quite easy to prove monotonicity of sensitivity with respect to K_{BM} . Consider an increase in K_{BM} ; we want to prove that $[PMT]$ necessarily decreases. For contradiction, assume not ($[PMT] \geq [PMT_{old}]$). Thus by the mass action formula for PMT formation, $[MT] > [MT_{old}]$ since K_{PM} unchanged and $[Probe]$ decreases. Thus by the conservation equation $[MT] + [PMT] + [BMT] = c_0$, $[BMT] < [BMT_{old}]$. But $[BMT] < [BMT_{old}]$ ($\Rightarrow [Blocker] > [Blocker_{old}]$) and $[MT] > [MT_{old}]$ violates the mass action law for BMT formation, since K_{BM} has increased. Thus we have a contradiction and $[PMT]$ necessarily decreases. It may be possible to make progress along the other lines of investigation using similar reasoning.

Response: We thank the reviewer for providing a brilliant and concise demonstration of the monotonicity of sensitivity and $[PMT]$ over K_{BM} . We have added the proof by contradiction in the revised SI (line 101-125). Also, using similar reasoning, we demonstrated that the specificity of the dissociative 4-Way SELECT system was monotonically increasing over K_{BW} (line 405-427 in the revised SI).

3. Variables appear without definition, or are defined without appearing, in numerous places. See, for example:

- R and T on line 101

- $[Ps]_0$. Is this the total concentration of the strand, or the initial concentration of free strand?

- The monotonically increasing/decreasing arrows nwws to be defined. In addition, in eq. 12 and 13 it doesn't even say which quantity they increase or decrease with respect to.

- The K and X expression on line 283.

Response: We have revised the above points in the revised manuscript. We also checked the whole manuscript and revised similar mistakes. $[Ps]_0$ (named as $[C]_0$ in the revised manuscript) is the initial concentration of free strand (we have explained this on line 193 in the revised manuscript).

4. The "SELECT" name is a big improvement, but the systems described on lines 165 and 182 need to be more clearly distinguished.

Response: To make it clearer, we named the system of Figure 1 as "Standard probe/standard blocker composition system", the system of Figure S1 as "Strand displacement probe/strand displacement blocker composition system" and the system of Figure 2 as "Strand displacement probe/standard blocker composition system".

5. $F(x)$ is an important function. It would make sense to write down the equations from which it follows. I think it would be better to explicitly describe it as a function of the concentrations as well as K (or x). This would really help, since the concentrations that go into F are different in different contexts.

Response: We have defined the function of F as a binary function of $F(x, [PC]_0)$ (line 192 in the revised manuscript). We also described the detailed derivation process of $F(x, [PC]_0)$ in the revised SI (line 271-278). In the experiments that $[PC]_0$ remained constant while the blocker sequence and concentration varied, we simplified the binary function of $F(x, [PC]_0)$ as a unary function of $F(x)$ to make the mathematics more concise (line 197-199 in the revised manuscript).

6. What does quasi-monotonically mean?

Response: Quasi-monotonically increasing means that although under certain situations the specificity was not rigorously increasing over $[B]_0$ and $-\Delta G_{BW}$, and the specificity curve could present small fluctuations, the amplitude of the fluctuations was so tiny that as a whole, the specificity seemed to be increasing over $[B]_0$ and $-\Delta G_{BW}$. To make it more understandable, we deleted the phrase "quasi-monotonically" and described the variation of specificity in a more detailed way in the revised manuscript (line 202-205)

7. I think "decreasing" on line 269 should be "increasing".

Response: Yes, it should be "increasing", and we have corrected it in the revised manuscript.

8. Personally, I think it would REALLY help if all strands were given a symbol, and complexes were represented by putting the two symbols together (eg. P, MT and PMT). The current system is confusing.

Response: This is a very good suggestion, and we have given all strands a symbol in the revised manuscript and SI.

9. On the assumption of $[Blocker]=[Blocker]-c_0$.

(a) This is actually a different sort of assumption in different cases. In the original system, it is essentially an assumption that either blocker is more prevalent, or binds more strongly to the target strand, than the probe. It is not, as the authors say in 435, an assumption that the blocker strand "changes very little". It changes as much as possible!

(b) In the SELECT context, the assumption is satisfied immediately if bonding of blocker to target or Ps is strong. There is no requirement that the blocker out compete the probe. It is likely to be an extremely good assumption in this setting.

(c) What do the authors mean by "in the beginning" on line 441? It sounds, from the previous sentence, like they are talking about the beginning of the experiment, but this wouldn't make sense.

Response: We thank the reviewer for pointing out the subtle point of the assumption for our SELECT system. We have discussed the requirement of this assumption for conventional

competitive systems and our proposed 4-Way SELECT system respectively in the revised manuscript (line 429-434).

We used the phrase “in the beginning” to represent the early stage of the process of changing $[B]_0$ or $-\Delta G_{BW}$ from low levels to high levels, i.e. at this stage, $[B]_0$ and $-\Delta G_{BW}$ were small. To make it more understandable, we deleted the phrase and directly indicated the levels of $[B]_0$ and $-\Delta G_{BW}$ in the revised manuscript (line 434-440).

Reviewers' Comments:

Reviewer #1:

Remarks to the Author:

I am pleased with the authors' changes, and would now recommend publication. Minor comments:

The authors define the arrows as "monotonically increasing or decreasing", but don't say with respect to what. Increase of any input parameter?

The notation has improved vastly, but are $[C_0]$ and c_0 now a little problematic?

Minor fluctuations in the specificity? I'd have thought the curve would be fairly smooth. Are the authors sure that this isn't numerical error?

the authors are now labelling the transitions with dG^0 , and have a gap related to the size of dG^0 , in the figure. This is good. I'm therefore a little uncertain about the meaning of the scale bar in the centre of the graph?

Responses to the reviewers

We would like to thank the reviewer for the valuable comments and suggestions, and we have carefully revised our manuscript with all changes tracked. The point-by-point responses to each comment are as follows:

I am pleased with the authors' changes, and would now recommend publication.

Minor comments:

1. The authors define the arrows as "monotonically increasing or decreasing", but don't say with respect to what. Increase of any input parameter?

Response: We have defined the arrows as "the function was monotonically increasing or decreasing with its variables" in the revised manuscript (Line 111 and 113).

2. The notation has improved vastly, but are [C₀] and c₀ now a little problematic?

Response: We have changed the notation of [C] to [S] in the revised manuscript.

3. Minor fluctuations in the specificity? I'd have thought the curve would be fairly smooth. Are the authors sure that this isn't numerical error?

Response: In the previous manuscript and supplementary information, we have explained that the function of specificity was not always monotonically increasing over [B]₀ and $-\Delta G_{BW}$ under all occasions. For certain designs of the probes and the corresponding thermodynamic parameters, the derivative of the function of specificity was not always positive when varying [B]₀ and $-\Delta G_{BW}$, and thereby the curve could present small fluctuations.

However, regarding the specificity curve of Figure 2d, the thermodynamic parameters of probe-3 were appropriate and the specificity was rigorously increasing over [B]₀ and $-\Delta G_{BW}$, so the curve was fairly smooth.

To make it clearer, we have added the in the revised main article that "It was worth noting that the thermodynamic parameters of probe-3 were appropriate and the specificity was rigorously increasing over [B]₀ and $-\Delta G_{BW}$ " (Line 202-204).

4. The authors are now labelling the transitions with dG^0 , and have a gap related to the size of dG^0 , in the figure. This is good. I'm therefore a little uncertain about the meaning of the scale bar in the centre of the graph?

Response: As was pointed out by the reviewer, when we drew the reaction pathways, we made the sizes of the gaps related to the values of ΔG . However, only by the sizes of the gaps, it is quite difficult to get a whole picture of the levels and relations of all associated free energy changes. Since the whole picture is very helpful for understanding the working mechanism of the 4-Way SELECT system, we chose to retain the scale bar in the centre of the graph to directly show the levels and relations of all associated free energy changes.